# The Inductive Bias of Flatness Regularization for Deep Matrix Factorization

**Khashayar Gatmiry**
MIT
gatmiry@mit.edu

Zhiyuan Li
Stanford University
zhiyuanli@stanford.edu

Ching-Yao Chuang
MIT
cychuang@mit.edu

Sashank Reddi
Google
sashank@google.com

Tengyu Ma
Stanford University
tengyuma@stanford.edu

Stefanie Jegelka
TU Munich & MIT
stefje@csail.mit.edu

## Abstract

Recent works on over-parameterized neural networks have shown that the stochasticity in optimizers has the implicit regularization effect of minimizing the sharpness of the loss function (in particular, the trace of its Hessian) over the family zero-loss solutions. More explicit forms of flatness regularization also empirically improve the generalization performance. However, it remains unclear why and when flatness regularization leads to better generalization. This work takes the first step toward understanding the inductive bias of the minimum trace of the Hessian solutions in an important setting: learning deep linear networks from linear measurements, also known as *deep matrix factorization*. We show that for all depth greater than one, with the standard Restricted Isometry Property (RIP) on the measurements, minimizing the trace of Hessian is approximately equivalent to minimizing the Schatten 1-norm of the corresponding end-to-end matrix parameters (i.e., the product of all layer matrices), which in turn leads to better generalization. We empirically verify our theoretical findings on synthetic datasets.

## 1 Introduction

Modern deep neural networks are typically over-parametrized and equipped with huge model capacity, but surprisingly, they generalize well when trained using stochastic gradient descent (SGD) or its variants [51]. A recent line of research suggested the *implicit bias* of SGD as a possible explanation to this mysterious ability. In particular, Damian et al. [10], Li et al. [29], Arora et al. [4], Lyu et al. [33], Wen et al. [48], Liu et al. [31] have shown that SGD can implicitly minimize the *sharpness* of the training loss, in particular, the trace of the Hessian of the training loss, to obtain the final model. However, despite the strong empirical evidence on the correlation between various notions of sharpness and generalization [25, 23, 38, 24] and the effectiveness of using sharpness regularization on improving generalization [16, 49, 53, 39], the connection between penalization of the sharpness of training loss and better generalization still remains majorly unclear [13, 2] and has only been proved in the context of two-layer linear models [29, 37, 12]. To further understand this connection beyond the two layer case, we study the inductive bias of penalizing the *trace of the Hessian* of training loss and its effect on the *generalization* in an important theoretical deep learning setting: *deep linear networks* (or equivalently, *deep matrix factorization* [3]). We start by briefly describing the problem setup.

**Deep Matrix Factorization.** Consider an $L$-layer deep network where $L \in \mathbb{N}^+, L \geq 2$ is the depth of the model. Let $W_i \in \mathbb{R}^{d_i \times d_{i-1}}$ and $d_i$ denote the layer weight matrix and width of the $i^{\text{th}}$ ($i \in [L]$)

layer respectively. We use $\mathbf{W}$ to denote the concatenation of all the parameters $(W_1, \ldots, W_L)$ and define the *end-to-end matrix* of $\mathbf{W}$ as

$$E(\mathbf{W}) \triangleq W_L W_{L-1} \cdots W_1. \tag{1}$$

In this paper, we focus on models that are linear in the space of the end-to-end matrix $E(W)$. Suppose $M^* \in \mathbb{R}^{d_L \times d_0}$ is the target end-to-end matrix, and we observe $n$ linear measurements (matrices) $A_i \in \mathbb{R}^{d_L \times d_0}$ and the corresponding labels $b_i = \langle A_i, M^* \rangle$. The training loss of $\mathbf{W}$ is the mean-squared error (MSE) between the prediction $\langle A_i, W_L W_{L-1} \cdots W_1 \rangle$ and the observation $b_i$:

$$\mathcal{L}(\mathbf{W}) \triangleq \frac{1}{n} \sum_{i=1}^{n} \left( \langle A_i, W_L W_{L-1} \cdots W_1 \rangle - b_i \right)^2. \tag{2}$$

Throughout this paper, we assume that $d_i \geq \min(d_0, d_L)$ for each $i \in [L]$ and, thus, the image of the function $E(\cdot)$ is the entire $\mathbb{R}^{d_L \times d_0}$. In particular, this ensures that the deep models are sufficiently expressive in the sense that $\min_{\mathbf{W}} \mathcal{L}(\mathbf{W}) = 0$. For this setting, we aim to understand the structure of the trace of the Hessian minimization, as described below. The trace of Hessian is the sum of the eigenvalues of Hessian, which is an indicator of sharpness and it is known that variants of SGD, such as label noise SGD or 1-SAM, are biased toward models with a smaller trace of Hessian [29, 48].

**Min Trace of Hessian Interpolating Solution.** Our primary object of study is the interpolating solution with the minimum trace of Hessian, defined as:

$$\mathbf{W}^* \in \operatorname*{arg\,min}_{\mathbf{W}: \mathcal{L}(\mathbf{W})=0} \operatorname{tr}[\nabla^2 \mathcal{L}(\mathbf{W})]. \tag{3}$$

As we shall see shortly, the solution to the above optimization problem is not unique. We are interested in understanding the underlying structure of any minimizer $\mathbf{W}^*$. This will, in turn, inform us about the generalization nature of these solutions.

## 1.1 Main Results

Before delving into the technical details, we state our main results in this section. This also serves the purpose of highlighting the primary technical contributions of the paper. First, since the generalization of $\mathbf{W}$ only depends on its end-to-end matrix $E(\mathbf{W})$, it is informative to derive the properties of $E(\mathbf{W}^*)$ for any min trace of the Hessian interpolating solution $\mathbf{W}^*$ defined in (3). Indeed, penalizing the trace of Hessian in the $W$ space induces an equivalent penalization in the space of the end-to-end parameters. More concretely, given an end-to-end parameter $M$, let the induced regularizer $F(M)$ denote the minimum trace of Hessian of the training loss at $\mathbf{W}$ among all $\mathbf{W}$'s that instantiate the end-to-end matrix $M$ i.e., $E(\mathbf{W}) = M$.

**Definition 1** (Induced Regularizer). *Suppose $M \in \mathbb{R}^{d_L \times d_0}$ is an end-to-end parameter that fits the training data perfectly (that is, $\langle A_i, M \rangle = b_i, \ \forall i \in [n]$). We define the* induced regularizer *as*

$$F(M) \triangleq \min_{\mathbf{W}: E(\mathbf{W})=M} \operatorname{tr}[\nabla^2 \mathcal{L}(\mathbf{W})] \tag{4}$$

Since the image of $E(\cdot)$ is the entire $\mathbb{R}^{d_L \times d_0}$ by our assumption that $d_i \geq \min(d_0, d_L)$, function $F$ is well-defined for all $M \in \mathbb{R}^{d_L \times d_0}$. It is easy to see that minimizing the trace of the Hessian in the original parameter space (see (3)) is equivalent to penalizing $F(M)$ in the end-to-end parameter. Indeed, the minimizers of the implicit regularizer in the end-to-end space are related to the minimizers of the implicit regularizer in the $W$ space, i.e.,

$$\operatorname*{arg\,min}_{M: \mathcal{L}'(M)=0} F(M) = \left\{ E(\mathbf{W}^*) \mid \mathbf{W}^* \in \operatorname*{arg\,min}_{\mathbf{W}: \mathcal{L}(\mathbf{W})=0} \operatorname{tr}[\nabla^2 \mathcal{L}(\mathbf{W})] \right\},$$

where for any $M \in \mathbb{R}^{d_L \times d_0}$, we define $\mathcal{L}'(M) \triangleq \frac{1}{n} \sum_{i=1}^{n} \left( \langle A_i, M \rangle - b_i \right)^2$ and thus $\mathcal{L}(\mathbf{W}) = \mathcal{L}'(E(\mathbf{W}))$. This directly follows from the definition of $F$ in (4). Our main result characterizes the induced regularizer $F(M)$ when the data satisfies the RIP property.

**Theorem 1** (Induced regularizer under RIP). *Suppose the linear measurements $\{A_i\}_{i=1}^{n}$ satisfy the $(1, \delta)$-RIP condition.*

| Settings | Induced Regularizer $F(M)/L$ | Theorem |
|---|---|---|
| $(1,\delta)$-RIP | $(1 \pm O(\delta))(d_0 d_L)^{1/L}\|M\|_*^{2-2/L}$ | Theorem 1 |
| $L = 2$ | $\left\|\left(\frac{1}{n}A_i A_i^\top\right)^{1/2} M \left(\frac{1}{n}A_i^\top A_i\right)^{1/2}\right\|_*$ | Theorem 5 ([12]) |
| $n = 1$ | $\left\|\left(A^T M\right)^{L-1} A^T\right\|_{S_{2/L}}^{2/L}$ | Theorem 7 |

Table 1: Summary of properties of the induced regularizer in the end-to-end matrix space. Here $\|\cdot\|_{S_p}$ denotes the Schatten $p$-norm for $p \in [1,\infty]$ and Schatten $p$-quasinorm for $p \in (0,1)$ (see Definition 2). $\|\cdot\|_*$ denotes the Schatten 1-norm, also known as the nuclear norm.

1. *For any $M \in \mathbb{R}^{d_L \times d_0}$ such that $\langle A_i, M \rangle = b_i$, $\forall i \in [n]$, it holds that*

$$(1-\delta)L(d_0 d_L)^{1/L}\|M\|_*^{2(L-1)/L} \le F(M) \le (1+\delta)L(d_0 d_L)^{1/L}\|M\|_*^{2(L-1)/L}. \quad (5)$$

2. *Let $\boldsymbol{W}^* \in \arg\min_{\boldsymbol{W}:\mathcal{L}(\boldsymbol{W})=0} \operatorname{tr}[\nabla^2 \mathcal{L}(\boldsymbol{W})]$ be an interpolating solution with minimal trace of Hessian . Then $E(\boldsymbol{W}^*)$ roughly minimizes the nuclear norm among all interpolating solutions of $\mathcal{L}'$. That is,*

$$\|E(\boldsymbol{W}^*)\|_* \le \frac{1+\delta}{1-\delta} \min_{\mathcal{L}'(M)=0} \|M\|_*.$$

However, for more general cases, it is challenging to compute the closed-form expression of $F$. In this work, we derive closed-form expressions for $F$ in the following two cases: (1) depth $L$ is equal to 2 and (2) there is only one measurement, *i.e.*, $n = 1$ (see Table 1). Leveraging the above characterization of induced regularzier, we obtain the following result on the generalization bounds:

**Theorem 2** (Recovery of the ground truth under RIP). *Suppose the linear measurements $\{(A_i)\}_{i=1}^n$ satisfy the $(2, \delta(n))$-RIP (Definition 3). Then for any $\boldsymbol{W}^* \in \arg\min_{\boldsymbol{W}:\mathcal{L}(\boldsymbol{W})=0} \operatorname{tr}[\nabla^2 \mathcal{L}(\boldsymbol{W})]$, we have*

$$\|E(\boldsymbol{W}^*) - M^*\|_F^2 \le \frac{8\delta(n)}{(1-\delta(n))^2}\|M^*\|_*^2. \quad (6)$$

*where $\delta(n)$ depends on the number of measurements $n$ and the distribution of the measurements.*

If we further suppose $\{A_i\}_{i=1}^n$ are independently sampled from some distribution over $\mathbb{R}^{d_L \times d_0}$ satisfying that $\mathbb{E}_A \langle A, M \rangle^2 = \|M\|_F^2$, *e.g.*, the standard multivariate Gaussian distribution, denoted by $\mathcal{G}_{d_L \times d_0}$, we know $\delta(n) = O(\sqrt{\frac{d_L + d_0}{n}})$ from Candes and Plan [7] (see Section 5.1 for more examples).

**Theorem 3.** *For $n \ge \Omega(r(d_0 + d_L))$, with probability at least $1 - \exp(\Omega(d_0 + d_L))$ over the randomly sampled $\{A_i\}_{i=1}^n$ from multivariate Gaussian distribution $\mathcal{G}$, for any minimum trace of Hessian interpolating solution $\boldsymbol{W}^* \in \arg\min_{\boldsymbol{W}:\mathcal{L}(\boldsymbol{W})=0} \operatorname{tr}[\nabla^2 \mathcal{L}(\boldsymbol{W})]$, the population loss $\overline{\mathcal{L}}(\boldsymbol{W}^*) \triangleq \mathbb{E}_{A \sim \mathcal{G}}(\langle A, E(\boldsymbol{W}^*) \rangle - \langle A, M^* \rangle)^2$ satisfies that*

$$\overline{\mathcal{L}}(\boldsymbol{W}^*) = \|E(\boldsymbol{W}^*) - M^*\|_F^2 \le O\left(\frac{d_0 + d_L}{n}\|M^*\|_*^2 \log^3 n\right).$$

Next, we state a lower bound for the conventional estimator for overparameterized models that minimizes the norm. The lower bound states that, to achieve a small error, the number of samples should be as large as the product of the dimensions of the end-to-end matrix $d_0 d_L$ as opposed to $d_0 + d_L$ in case of the min trace of Hessian minimizer. It is proved in Appendix F.

**Theorem 4** (Lower bound for $\ell_2$ regression). *Suppose $\{A_i\}_{i=1}^n$ are randomly sampled from multivariate Gaussian distribution $\mathcal{G}$, let $\tilde{\boldsymbol{W}} = \arg\min_{\boldsymbol{W}:\mathcal{L}(\boldsymbol{W})=0} \|E(\boldsymbol{W})\|_F$ to be the minimum Frobenius norm interpolating solution, then the expected population loss is*

$$\mathbb{E}\,\overline{\mathcal{L}}(\tilde{\boldsymbol{W}}) = (1 - \frac{\min\{n, d_0 d_L\}}{d_0 d_L})\|M^*\|_F^2.$$

The lower bound in Theorem 4 shows in order to obtain an $O(1)$-relatively accurate estimates of the ground truth in expectation, namely to guarantee $\mathbb{E}\,\overline{\mathcal{L}}(\tilde{W}) \leq O(1)\|M^*\|_F^2$, the minimum Frobenius norm interpolating solution needs at least $\Omega(d_0 d_L)$ samples. In contrast, the minimizer of trace of Hessian in the same problem only requires $O((d_0 + d_L)\|M^*\|_*^2/\|M^*\|_F^2)$ samples, which is at least $\tilde{O}(\frac{\min\{d_0, d_L\}}{r})$ times smaller. We further illustrate experimentally the superior generalization ability of sharpness minimization algorithms like label noise SGD [6, 10, 29] compared to vanilla mini-batch SGD Figure 1. Due to the space limits, we defer the full setting for experiments into Appendix A.

## 2    Related Work

**Connection Between Sharpness and Generalization.**    Research on the connection between generalization and sharpness dates back to Hochreiter and Schmidhuber [21]. Keskar et al. [25] famously observe that when increasing the batch size of SGD, the test error and the sharpness of the learned solution both increase. Jastrzebski et al. [23] extend this observation and found that there is a positive correlation between sharpness and the ratio between learning rate and batch size. Jiang et al. [24] perform a large-scale empirical study on various notions of generalization measures and show that sharpness-based measures correlate with generalization best. Liu et al. [31] find that among language models with the same validation pretraining loss, those that have smaller sharpness can have better downstream performance. On the other hand, Dinh et al. [13] argue that for networks with scaling invariance, there always exist models with good generalization but with arbitrarily large sharpness. We note this does not contradict our main result here, which only asserts the interpolation solution with a minimal trace of Hessian generalizes well, but not vice versa. Empirically, sharpness minimization is also a popular and effective regularization method for overparametrized models [39, 17, 53, 49, 26, 32, 54, 52, 1].

**Implicit Bias of Sharpness Minimization.**    Recent theoretical works [6, 10, 29, 31] show that SGD with label noise is implicitly biased toward local minimizers with a smaller trace of Hessian under the assumption that the minimizers locally connect as a manifold. Such a manifold setting is empirically verified by Draxler et al. [14], Garipov et al. [18] in the sense that the set of minimizers of the training loss is path-connected. It is the same situation for the deep matrix factorization problem studied in this paper, although we do not study the optimization trajectory. Instead, we directly study properties of the minimum trace of Hessian interpolation solution.

Sharpness-reduction implicit bias can also happen for deterministic GD. Arora et al. [4] show that normalized GD implicitly penalizes the largest eigenvalue of the Hessian. Ma et al. [35] argues that such sharpness reduction phenomena can also be caused by a multi-scale loss landscape. Lyu et al. [33] show that GD with weight decay on a scale-invariant loss function implicitly decreases the spherical sharpness, *i.e.*, the largest eigenvalue of the Hessian evaluated at the normalized parameter. Another line of work focuses on the sharpness minimization effect of a large learning rate in GD, assuming that it converges at the end of training. This has been studied mainly through linear stability analysis [50, 8, 34, 9]. Recent theoretical analysis [11, 30] showed that the sharpness minimization effect of a large learning rate in GD does not necessarily rely on convergence and linear stability, through a four-phase characterization of the dynamics at the Edge of Stability regime [8].

## 3    Preliminaries

**Notation.** We use $[n]$ to denote $\{1, 2, \ldots, n\}$ for every $n \in \mathbb{N}$. We use $\|M\|_F$, $\|M\|_*$, $\|M\|_2$ and $\mathrm{tr}(M)$ to denote the Frobenius norm, nuclear norm, spectral norm and trace of matrix $M$ respectively. For any function $f$ defined over set $S$ such that $\min_{x \in S} f(x)$ exists, we use $\arg\min_S f$ to denote the set $\{y \in S \mid f(y) = \min_{x \in S} f(x)\}$. Given a matrix $M$, we use $h_M$ to denote the linear map $A \mapsto \langle A, M \rangle$. We use $\mathcal{H}_r$ to to denote the set $\mathcal{H}_r \triangleq \{h_M \mid \|M\|_* \leq r\}$. $M_{i:}$ and $M_{:j}$ are used to denote the $i$th row and $j$th column of the matrix $M$.

The following definitions will be important to the technical discussion in the paper.

**Rademacher Complexity.** Given $n$ data points $\{A_i\}_{i=1}^n$, the *empirical Rademacher complexity* of function class $\mathcal{H}$ is defined as

$$\mathcal{R}_n(\mathcal{H}) = \frac{1}{n} \mathbb{E}_{\epsilon \sim \{\pm 1\}^n} \sup_{h \in \mathcal{H}} \sum_{i=1}^n \epsilon_i h(A_i).$$

Given a distribution $P$, the *population Rademacher complexity* is defined as follows: $\overline{\mathcal{R}}_n(\mathcal{H}) = \mathbb{E}_{A_i \overset{iid}{\sim} P} \mathcal{R}_n(\mathcal{H})$. This is mainly used to upper bound the generalization gap of SGD.

**Definition 2** (Schatten $p$-(quasi)norm). *Given any $d, d' \in \mathbb{N}^+$, $p \in (0, \infty)$ a matrix $M \in \mathbb{R}^{d \times d'}$ with singular values $\sigma_1(M), \ldots, \sigma_{\min(d,d')}(M)$, we define the Schattern $p$-(semi)norm as*

$$\|M\|_{S_p} = \left( \sum_{i=1}^{\min(d,d')} \sigma_i^p(M) \right)^{1/p}.$$

Note that in this definition $\|\cdot\|_{S_p}$ is a norm only when $p \geq 1$. When $p \in (0, 1)$, the triangle inequality does not hold. Note that when $p \in (0, 1)$, $\|A + B\|_{S_p} \leq 2^{1/p-1}(\|A\|_{S_p} + \|B\|_{S_p})$ for any matrices $A$ and $B$, however, $2^{1/p-1} > 1$.

We use $L$ to denote the depth of the linear model and $\boldsymbol{W} = (W_1, \ldots, W_L)$ to denote the parameters, where $W_i \in \mathbb{R}^{d_i \times d_{i-1}}$. We assume that $d_i \geq \min(d_0, d_L)$ for each $i \in [L-1]$ and, thus, the image of $E(\boldsymbol{W})$ is the entire $\mathbb{R}^{d_L \times d_0}$. Following is a simple relationship between nuclear norm and Frobenius norm that is used frequently in the paper.

**Lemma 1.** *For any matrices $A$ and $B$, it holds that $\|AB\|_* \leq \|A\|_F \|B\|_F$.*

# 4 Exact Formulation of Induced Regularizer by Trace of Hessian

In this section, we derive the exact formulation of trace of Hessian for $\ell_2$ loss over deep matrix factorization models with linear measurements as a minimization problem over $\boldsymbol{W}$. We shall later approximate this formula by a different function in Section 5, which allows us to calculate the implicit bias in closed-form in the space of end-to-end matrices.

We first introduce the following simple lemma showing that the trace of the Hessian of the loss is equal to the sum of squares of norms of the gradients of the neural network output.

**Lemma 2.** *For any twice-differentiable function $\{f_i(\boldsymbol{W})\}_{i=1}^n$, real-valued labels $\{b_i\}_{i=1}^n$, loss function $\mathcal{L}(\boldsymbol{W}) = \frac{1}{n} \sum_{i=1}^n (f_i(\boldsymbol{W}) - b_i)^2$, and any $\boldsymbol{W}$ satisfying $\mathcal{L}(\boldsymbol{W}) = 0$, it holds that*

$$\mathrm{tr}(\nabla^2 \mathcal{L}(\mathbf{W})) = \frac{2}{n} \sum_{i=1}^n \|\nabla f_i(\mathbf{W})\|^2.$$

Using Lemma 2, we calculate the trace of Hessian for the particular loss defined in (2). To do this, we consider $\mathbf{W}$ in Lemma 2 to be the concatenation of matrices $(W_1, \ldots, W_L)$ and we set $f_i(\boldsymbol{W})$ to be the linear measurement $\langle A_i, E(\boldsymbol{W}) \rangle$, where $E(\boldsymbol{W}) = W_L \cdots W_1$ (see (1)). To calculate the trace of Hessian, according to Lemma 2, we need to calculate the gradient of $\mathcal{L}(\boldsymbol{W})$ in (2). To this end, for a fixed $i$, we compute the gradient of $\langle A_i, E(\boldsymbol{W}) \rangle$ with respect to one of the weight matrices $W_j$.

$$\begin{aligned}
\nabla_{W_j} \langle A_i, E(\boldsymbol{W}) \rangle &= \nabla_{W_j} \mathrm{tr}(A_i^\top W_L \ldots W_1) \\
&= \nabla_{W_j} \mathrm{tr}((W_{j-1} \ldots W_1 A_i^\top W_L \ldots W_{j+1}) W_j) \\
&= (W_{j-1} \ldots W_1 A_i^\top W_L \ldots W_{j+1})^\top.
\end{aligned}$$

According to Lemma 2, trace of Hessian is given by

$$\mathrm{tr}(\nabla^2 L)(\mathbf{W}) = \frac{1}{n} \sum_{i=1}^n \sum_{j=1}^L \|\nabla_{W_j} \langle A_i, E(\boldsymbol{W}) \rangle\|_F^2 = \frac{1}{n} \sum_{i=1}^n \sum_{j=1}^L \|W_{j-1} \ldots W_1 A_i^\top W_L \ldots W_{j+1}\|_F^2.$$

As mentioned earlier, our approach is to characterize the minimizer of the trace of Hessian among all interpolating solutions by its induced regularizer in the end-to-end matrix space. The above

calculation provides the following more tractable characterization of induced regularizer $F$ in (12):

$$F(M) = \min_{E(\boldsymbol{W})=M} \sum_{i=1}^{n} \sum_{j=1}^{L} \|W_{j-1} \dots W_1 A_i^\top W_L \dots W_{j+1}\|_F^2. \tag{7}$$

In general, we cannot solve $F$ in closed form for general linear measurements $\{A_i\}_{i=1}^{n}$; however, interestingly, we show that it can be solved approximately under reasonable assumption on the measurements. In particular, we show that the induced regularizer, as defined in (7), will be approximately proportional to a power of the nuclear norm of $E(\boldsymbol{W})$ given that the measurements $\{A_i\}_{i=1}^{n}$ satisfy a natural norm-preserving property known as the Restricted Isometry Property (RIP) [7, 42].

Before diving into the proof of the general result for RIP, we first illustrate the connection between nuclear norm and the induced regularizer for the depth-two case. In this case, fortunately, we can compute the closed form of the induced regularizer. This result was first proved by Ding et al. [12]. For self-completeness, we also provide a short proof.

**Theorem 5** (Ding et al. [12]). *For any $M \in \mathbb{R}^{d_L \times d_0}$, it holds that*

$$F(M) \triangleq \min_{W_2 W_1 = M} \operatorname{tr}[\nabla^2 \mathcal{L}](\boldsymbol{W}) = 2 \left\| \left( \tfrac{1}{n} \sum_i A_i A_i^\top \right)^{1/2} M \left( \tfrac{1}{n} \sum_i A_i^\top A_i \right)^{1/2} \right\|_*. \tag{8}$$

*Proof of Theorem 5.* We first define $B_1 = (\sum_{i=1}^{n} A_i A_i^T)^{\frac{1}{2}}$ and $B_2 = (\sum_{i=1}^{n} A_i^T A_i)^{\frac{1}{2}}$. Therefore we have that

$$\operatorname{tr}[\nabla^2 \mathcal{L}](\boldsymbol{W}) = \sum_{i=1}^{n} \left( \|A_i^T W_2\|_F^2 + \|W_1 A_i^T\|_F^2 \right) = \|B_1 W_2\|_F^2 + \|W_1 B_2\|_F^2.$$

Further applying Lemma 1, we have that

$$F(M) = \min_{W_2 W_1 = M} \operatorname{tr}[\nabla^2 \mathcal{L}](\boldsymbol{W}) = \min_{W_2 W_1 = M} \sum_{i=1}^{n} \left( \|A_i^T W_2\|_F^2 + \|W_1 A_i^T\|_F^2 \right)$$

$$\geq \min_{W_2 W_1 = M} 2\|B_1 W_2 W_1 B_2\|_*^2 = 2\|B_1 M B_2\|_*^2.$$

Next we show this lower bound of $F(M)$ can be attained. Let $U\Lambda V^T$ be the SVD of $B_1 M B_2$. The equality condition happens for $W_2^* = B_1^\dagger U \Lambda^{1/2}, W_1^* = \Lambda^{1/2} V^T B_2^\dagger$, where we have that $\sum_{i=1}^{n} \|A_i^T W_2^*\|_F^2 + \|W_1^* A_i^T\|_F^2 = 2\|\Lambda\|_F^2 = 2\|B_1 M B_2\|_F^2$. This completes the proof. □

The right-hand side in (8) will be very close to the nuclear norm of $M$ if the two extra multiplicative terms are close to the identity matrix. It turns out that $\{A_i\}_{i=1}^{n}$ satisfying the $(1, \delta)$-RIP exactly guarantees the two extra terms are $O(\delta)$-close to identity. However, the case for deep networks where depth is larger than two is fundamentally different from the two-layer case, where one can obtain a closed form for $F$. To the best of our knowledge, it is open whether one obtain a closed form for the induced-regularizer for the trace of Hessian when $L > 2$. Nonetheless, in Section 5.1, we show that under RIP, we can still approximate it with the nuclear norm.

## 5 Results for Measurements with Restricted Isometry Property (RIP)

In this section, we present our main results for the generalization benefit of flatness regularization in deep linear networks. We structure the analysis as follows:

1. In Section 5.1, we first recap some preliminaries on the RIP property.

2. In Section 5.2, we prove that the induced regularizer by trace of Hessian is approximately the power of nuclear norm for $(1, \delta)$-RIP measurements (Theorem 1).

3. In Section 5.3, we prove that the minimum trace of Hessian interpolating solution with $(2, \delta)$-RIP measurements can recover the ground truth $M^*$ up to error $\delta \|M^*\|_*^2$. For $\{A_i\}_{i=1}^{n}$ sampled from Gaussian distributions, we know $\delta = O(\sqrt{\frac{d_0 + d_L}{n}})$.

4. In Section 5.4, we prove a generalization bound with faster rate of $\frac{d_0 + d_L}{n} \|M^*\|_*^2$ using local Rademacher complexity based techniques from Srebro et al. [44].

Next, we discuss important distributions of measurements for which the RIP property holds.

## 5.1 Preliminaries for RIP

**Definition 3** (Restricted Isometry Property (RIP)). *A family of matrices $\{A_i\}_{i=1}^n$ satisfies the $(r, \delta)$-RIP iff for any matrix $X$ with the same dimension and rank at most $r$:*

$$(1 - \delta)\|X\|_F^2 \leq \frac{1}{n}\sum_{i=1}^n \langle A_i, X\rangle^2 \leq (1 + \delta)\|X\|_F^2. \tag{9}$$

Next, we give two examples of distributions where $\Omega(r(d_0 + d_L))$ samples guarantee $(r, O(1))$-RIP. The proofs follow from Theorem 2.3 in [7].

**Example 1.** *Suppose for every $i \in \{1, \dots, n\}$, each entry in the matrix $A_i$ is an independent standard Gaussian random variable,* i.e., $A_i \overset{i.i.d.}{\sim} \mathcal{G}_{d_L \times d0}$. *For every constant $\delta \in (0, 1)$, if $n \geq \Omega(r(d_0 + d_L))$, then with probability $1 - e^{\Omega(n)}$, $\{A_i\}_{i=1}^n$ satisfies $(r, \delta)$-RIP.*

**Example 2.** *If each entry of $A_i$ is from a symmetric Bernoulli random variable with variance 1, i.e. for all $i, k, \ell$, entry $[A_i]_{k,\ell}$ is either equal to 1 or $-1$ with equal probabilities, then for any $r$ and $\delta$, $(r, \delta)$-RIP holds with same probability as in Example 1 if the same condition there is satisfied.*

## 5.2 Induced Regularizer of Trace of Hessian is Approximately Nuclear Norm

This section focuses primarily on the proof of Theorem 2. Our proof consists of two steps: (1) we show that the trace of Hessian of training loss at the minimizer $\boldsymbol{W}$ is multiplicatively $O(\delta)$-close to the regularizer $R(\boldsymbol{W})$ defined below (Lemma 3) and (2) we show that the induced regularizer of $R$, $F'(M)$, is proportional to $\|M\|_*^{2(L-1)/L}$ (Lemma 4).

$$R(\boldsymbol{W}) \triangleq \|W_L \dots W_2\|_F^2 d_0 + \sum_{j=2}^{L-1} \|W_L \dots W_{j+1}\|_F^2 \|W_{j-1} \dots W_1\|_F^2 + \|W_{L-1} \dots W_1\|_F^2 d_L. \tag{10}$$

**Lemma 3.** *Suppose the linear measurement $\{A_i\}_{i=1}^n$ satisfy $(1, \delta)$-RIP. Then, for any $\boldsymbol{W}$ such that $\mathcal{L}(\boldsymbol{W}) = 0$, it holds that*

$$(1 - \delta)R(\boldsymbol{W}) \leq \text{tr}(\nabla^2 L)(\boldsymbol{W}) \leq (1 + \delta)R(\boldsymbol{W}).$$

Since $\text{tr}(\nabla^2\mathcal{L})(\boldsymbol{W})$ closely approximates $R(\boldsymbol{W})$, we can study $R$ instead of $\text{tr}[\nabla^2\mathcal{L}]$ to understand the implicit bias up to a multiplicative factor $(1 + \delta)$. In particular, we want to solve the induced regularizer of $R(\boldsymbol{W})$ on the space of end-to-end matrices, $F'(M)$:

$$F'(M) \triangleq \min_{\boldsymbol{W}:W_L\cdots W_1=M} R(\boldsymbol{W}). \tag{11}$$

Surprisingly, we can solve this problem in closed form.

**Lemma 4.** *For any $M \in \mathbb{R}^{d_L \times d_0}$, it holds that*

$$F'(M) \triangleq \min_{\boldsymbol{W}:\ W_L\dots W_1=M} R(\boldsymbol{W}) = L(d_0 d_L)^{1/L}\|M\|_*^{2(L-1)/L}. \tag{12}$$

*Proof of Lemma 4.* Applying the $L$-version of the AM-GM to Equation (10):

$$(R(\mathbf{W})/L)^L \geq d_0\|W_L \cdots W_2\|_F^2 \cdot \|W_1\|_F^2 \|W_L \cdots W_3\|_F^2 \cdots \|W_{L-1} \cdots W_1\|_F^2 d_L. \tag{13}$$

$$= d_0 d_L \prod_{j=1}^{L-1}\left(\|W_L \cdots W_{j+1}\|_F^2 \|W_j \cdots W_1\|_F^2\right)$$

Now using Lemma 1, we have for every $1 \leq j \leq L - 1$:

$$\|W_L \dots W_{j+1}\|_F^2 \|W_j \dots W_1\|_F^2 \geq \|W_L \dots W_1\|_*^2 = \|M\|_*^2. \tag{14}$$

Multiplying Equation (14) for all $1 \leq j \leq L - 1$ and combining with Equation (13) implies

$$\min_{\{W|\ W_L\dots W_1=M\}} R(\mathbf{W}) \geq L(d_0 d_L)^{1/L}\|M\|_*^{2(L-1)/L}. \tag{15}$$

Now we show that equality can indeed be attained. To construct an example in which the equality happens, consider the singular value decomposition of $M$: $M = U\Lambda V^T$, where $\Lambda$ is a square matrix with dimension $\mathrm{rank}(M)$.

For $1 \leq i \leq L$, we pick $Q_i \in \mathbb{R}^{d_i \times \mathrm{rank}(M)}$ to be any matrix with orthonormal columns. Note that $\mathrm{rank}(M)$ is not larger than $d_i$ for all $1 \leq i \leq L$, hence such orthonormal matrices $Q_i$ exist. Then we define the following with $\alpha, \alpha' > 0$ being constants to be determined:

$$W_L = \alpha'\alpha^{-(L-2)/2}U\Lambda^{1/2}Q_{L-1}{}^T \in \mathbb{R}^{d_L \times d_{L-1}},$$
$$W_i = \alpha Q_i Q_{i-1}{}^T \in \mathbb{R}^{d_i \times d_{i-1}}, \quad \forall 2 \leq i \leq L-1,$$
$$W_1 = \alpha'^{-1}\alpha^{-(L-2)/2}Q_1\Lambda^{1/2}V^T \in \mathbb{R}^{d_1 \times d_0}.$$

Note that $\Lambda$ is a square matrix with dimension $\mathrm{rank}(M)$. First of all, note that the defined matrices satisfy

$$W_L W_{L-1}\ldots W_1 = \alpha^{L-2}\alpha^{-(L-2)}U\Lambda^{1/2}\Lambda^{1/2}V^T = M.$$

To gain some intuition, we check that the equality case for all the inequalities that we applied above. We set the value of $\alpha$ in a way that these equality cases can hold simultaneously. Note that for the matrix holder inequality that we applied in Equation (14):

$$\|W_L \ldots W_{j+1}\|_F^2 \|W_j \ldots W_1\|_F^2 = \|W_L \ldots W_1\|_*^2 = \|\Lambda^{1/2}\|_F^2,$$

independent of the choice of $\alpha$. It remains to check the equality case for the AM-GM inequality that we applied in Equation (13). We have for all $2 \leq j \leq L-1$:

$$\|W_L \ldots W_{j+1}\|_F \|W_{j-1} \ldots W_1\|_F$$
$$= \alpha^{j-2}\alpha^{-(L-2)/2}\alpha^{L-j-1}\alpha^{-(L-2)/2}\|U\Lambda^{1/2}\|_F\|\Lambda^{1/2}V^T\|_F = \alpha^{-1}\|\Lambda^{1/2}\|_F^2, \quad (16)$$

Hence, equality happens for all of them. Moreover, for cases $j = 1$ and $j = L$, we have

$$d_0\|W_L \ldots W_2\| = \|\Lambda^{1/2}\|_F d_0 \alpha'\alpha^{L-2}\alpha^{-(L-2)/2} = \|\Lambda^{1/2}\|_F d_0\alpha'\alpha^{(L-2)/2}. \quad (17)$$

$$d_L\|W_{L-1} \ldots W_1\| = \|\Lambda^{1/2}\|_F d_L \alpha'^{-1}\alpha^{L-2}\alpha^{-(L-2)/2} = \|\Lambda^{1/2}\|_F d_L\alpha'^{-1}\alpha^{(L-2)/2}. \quad (18)$$

Thus it suffices to set $\alpha' = (\frac{d_L}{d_0})^{1/2}$ and $\alpha = (\frac{\|\Lambda^{1/2}\|_F}{\sqrt{d_0 d_L}})^{2/L} = (\frac{\|M\|_*}{d_0 d_L})^{1/L}$ so that the left-hand sides of (16), (17), and (18) are equal, which implies that the lower bound in Equation (15) is actually an equality. The proof is complete. $\qquad\square$

Now we can prove Theorem 1 as an implication of Lemma 4.

*Proof of Theorem 1.* The first claim is a corollary of Lemma 3. We note that

$$F(M) = \min_{W_L \ldots W_1 = M} \mathrm{tr}[\nabla^2 \mathcal{L}](M) \leq (1+\delta) \min_{W_L \ldots W_1 = M} R(\mathbf{W}) = (1+\delta)F'(M)$$
$$F(M) = \min_{W_L \ldots W_1 = M} \mathrm{tr}[\nabla^2 \mathcal{L}](M) \geq (1-\delta) \min_{W_L \ldots W_1 = M} R(\mathbf{W}) = (1-\delta)F'(M).$$

For the second claim, pick $\bar{\mathbf{W}}$ that minimizes $R(\bar{\mathbf{W}})$ over all $\mathbf{W}$'s that satisfy the linear measurements, thus we have that

$$R(\bar{\mathbf{W}}) = L(d_0 d_L)^{1/L}\|E(\bar{\mathbf{W}})\|_*^{2(L-1)/L} = L(d_0 d_L)^{1/L}\min_{\mathcal{L}'(M)=0}\|M\|_*^{2(L-1)/L}. \quad (19)$$

Now from the definition of $E(\mathbf{W}^*)$,

$$\mathrm{tr}(\nabla^2 L)(\mathbf{W}^*) \leq \mathrm{tr}(\nabla^2 L)(\bar{\mathbf{W}}) \leq (1+\delta)R(\bar{\mathbf{W}}), \quad (20)$$

where the last inequality follows from the definition of $W$. On the other hand

$$\mathrm{tr}(\nabla^2 L)(\mathbf{W}^*) \geq (1-\delta)R(\bar{\mathbf{W}}) \geq (1-\delta)L(d_0 d_L)^{1/L}\|E(\mathbf{W}^*)\|_*^{2(L-1)/L}. \quad (21)$$

Combining (19), (20) and (21),

$$\|E(\mathbf{W}^*)\|_* \leq (\frac{1+\delta}{1-\delta})^{\frac{L}{2(L-1)}}\min_{\mathcal{L}'(M)=0}\|M\|_*.$$

The proof is completed by noting that $\frac{L}{2(L-1)} \leq 1$ for all $L \geq 2$. $\qquad\square$

Thus combining Example 1 and Theorem 1 with $\delta = 1/2$, we have the following corollary.

**Corollary 1.** *Let $\{A_i\}_{i=1}^n$ be sampled independently from Gaussian distribution $\mathcal{G}_{d_L \times d_0}$ where $n \geq \Omega((d_0 + d_L))$, with probability at least $1 - \exp(\Omega(n))$, we have*

$$\|E(\boldsymbol{W}^*)\|_* \leq 3 \min_{\mathcal{L}'(M)=0} \|M\|_* \leq 3 \|E(\boldsymbol{W}^*)\|_* .$$

## 5.3 Recovering the Ground truth

In this section, we prove Theorem 2. The idea is to show that under RIP, the empirical loss $\mathcal{L}(\boldsymbol{W})$ is a good approximation for the Frobenius distance of $E(\boldsymbol{W})$ to the ground truth $M^*$. To this end, we first introduce a very useful Lemma 5 below, whose proof is deferred to Appendix E.

**Lemma 5.** *Suppose the measurements $\{A_i\}_{i=1}^n$ satisfy the $(2, \delta)$-RIP condition. Then for any matrix $M \in \mathbb{R}^{d_L \times d_0}$, we have that*

$$\left| \frac{1}{n} \sum_{i=1}^n \langle A_i, M \rangle^2 - \|M\|_F^2 \right| \leq 2\delta \|M\|_*^2 .$$

We note that if $\{A_i\}_{i=1}^n$ are i.i.d. random matrices with each coordinate being independent, zero mean, and unit variance (like standard Gaussian distribution), then $\|W - M^*\|_F^2$ is the population squared loss corresponding to $W$. Thus, Theorem 2 implies a generalization bound for this case. Now we are ready to prove Theorem 2.

*Proof of Theorem 2.* Note that from Theorem 1,

$$\|E(\boldsymbol{W}^*)\|_* \leq \frac{1+\delta}{1-\delta} \min_{\mathcal{L}'(M)=0} \|M\|_* \leq \frac{1+\delta}{1-\delta} \|M^*\|_* ,$$

which implies the following by triangle inequality,

$$\|E(\boldsymbol{W}^*) - M^*\|_* \leq \|\tilde{E}(\boldsymbol{W}^*)\|_* + \|M^*\|_* \leq \frac{2}{1-\delta} \|M^*\|_* . \tag{22}$$

Combining (22) with Lemma 5 (with $M = E(\boldsymbol{W}^*) - M^*$):

$$\left| \frac{1}{n} \sum_{i=1}^n \langle A_i, E(\boldsymbol{W}^*) - M^* \rangle^2 - \|E(\boldsymbol{W}^*) - M^*\|_F^2 \right| \leq \frac{8\delta}{(1-\delta)^2} \|M^*\|_*^2 .$$

Since $W^*$ satisfies the linear constraints $\text{tr}(A_i E(\boldsymbol{W}^*)) = b_i$, $\frac{1}{n} \sum_{i=1}^n \langle A_i, E(\boldsymbol{W}^*) - M^* \rangle^2 = \frac{1}{n} \sum_{i=1}^n \left( \langle A_i, E(\boldsymbol{W}^*) \rangle - b_i \right)^2 = 0$, which completes the proof. $\qquad \square$

## 5.4 Generalization Bound

In this section, we prove the generalization bound in Theorem 3, which yields a faster rate of $O(\frac{d_0 + d_L}{n} \|M^*\|_*^2)$ compared to $O(\sqrt{\frac{d_0 + d_L}{n}} \|M^*\|_*^2)$ in Theorem 2. The intuition for this is as follows: By Corollary 1, we know that with very high probability, the learned solution has a bounded nuclear norm for its end-to-end matrix, no larger than $3 \|M^*\|_2$, where $M^*$ is the ground truth. The key mathematical tool is Theorem 6, which provides an upper bound on the population error of the learned interpolation solution that is proportional to the square of the Rademacher complexity of the function class $\mathcal{H}_{3\|M^*\|_*} = \{h_M \mid \|M\|_* \leq 3 \|M^*\|_*\}$.

**Theorem 6** (Theorem 1, Srebro et al. [44]). *Let $\mathcal{H}$ be a class of real-valued functions and $\ell : \mathbb{R} \times \mathbb{R} \to \mathbb{R}$ be a differentiable non-negative loss function satisfying that (1) for any fixed $y \in \mathbb{R}$, the partial derivative $\ell(\cdot, y)$ with respect to its first coordinate is $H$-Lipschitz and (2) $|\sup_{x,y} \ell(x,y)| \leq B$, where $H, B$ are some positive constants. Then for any $p > 0$, we have that with probability at least $1 - p$ over a random sample of size $n$, for any $h \in \mathcal{H}$ with zero training loss,*

$$\bar{\mathcal{L}}(h) \leq O\left( H \log^3 n \mathcal{R}_n^2(\mathcal{H}) + \frac{B \log(1/p)}{n} \right). \tag{23}$$

One technical difficulty is that Theorem 6 only works for bounded loss functions, but the $\ell_2$ loss on Gaussian data is unbounded. To circumvent this issue, we construct a smoothly truncated variant of $\ell_2$ loss (41) and apply Theorem 6 on that. Finally, we show that with a carefully chosen threshold, this truncation happens very rarely and, thus, does not change the population loss significantly. The proof can be found in Appendix E.

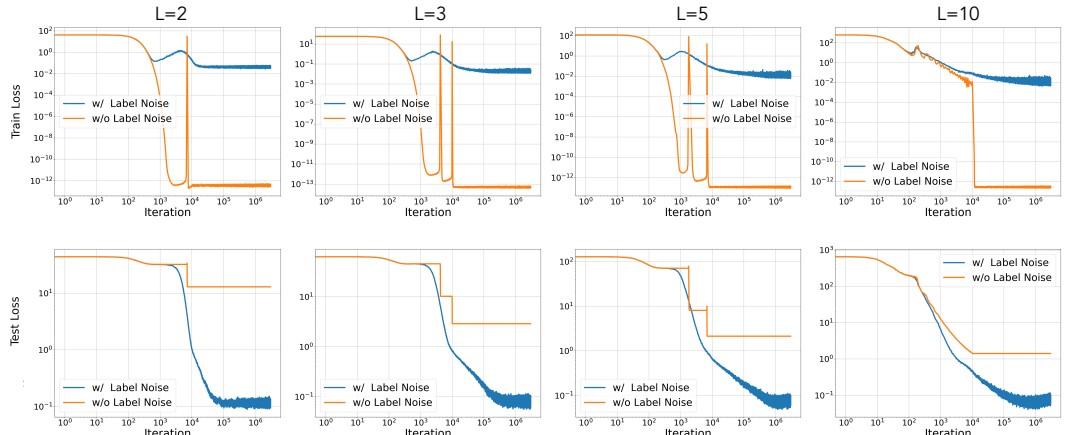

Figure 1: **Train and test loss.** Label noise SGD leads to better generalization results due to the sharpness-minimization implicit biases (as shown in Figure 2), while mini-batch SGD without label noise finds solutions with much larger test loss.

## 6   Result for the Single Measurement Case

Quite surprisingly, even though in the general case we cannot compute the closed-form of the induced regularizer in (12), we can find its minimum as a quasinorm function of the $E(\boldsymbol{W})$ which only depends on the singular values of $E(\boldsymbol{W})$. This yields the following result for multiple layers $L$ (possibly $L > 2$) with a single measurement.

**Theorem 7.** *Suppose there is only a single measurement matrix $A$, i.e., $n = 1$. For any $M \in \mathbb{R}^{d_L \times d_0}$, the following holds:*

$$F(M) = \min_{W_L \ldots W_1 = M} \mathrm{tr}[\nabla^2 \mathcal{L}](\boldsymbol{W}) = L \left\| \left(A^T M\right)^{L-1} A^T \right\|_{S_{2/L}}^{2/L}. \tag{24}$$

To better illustrate the behavior of this induced regularizer, consider the case where the measurement matrix $A$ is identity and $M$ is symmetric with eigenvalues $\{\sigma_i\}_{i=1}^d$. Then, it is easy to see that $F(M)$ in (24) is equal to $F(M) = \sum_i \sigma_i^{2(L-1)/L}$. Interestingly, we see that the value of $F(M)$ converges to the Frobenius norm of $M$ and not the nuclear norm as $L$ becomes large, which behaves quite differently (e.g. in the context of sparse recovery). This means that beyond RIP, the induced regularizer can behave very differently, and perhaps the success of training deep networks with SGD is closely tied to the properties of the dataset.

## 7   Conclusion and Future Directions

In this paper, we study the inductive bias of the minimum trace of the Hessian solutions for learning deep linear networks from linear measurements. We show that trace of Hessian regularization of loss on the end-to-end matrix of deep linear networks roughly corresponds to nuclear norm regularization under restricted isometry property (RIP) and yields a way to recover the ground truth matrix. Furthermore, leveraging this connection with the nuclear norm regularization, we show a generalization bound which yields a faster rate than Frobenius (or $\ell_2$ norm) regularizer for Gaussian distributions. Finally, going beyond RIP conditions, we obtain closed-form solutions for the case of a single measurement. Several avenues for future work remain open, e.g., more general characterization of trace of Hessian regularization beyond RIP settings and understanding it for neural networks with non-linear activations.

## Acknowledgement

TM and ZL would like to thank the support from NSF IIS 2045685. KG and SJ acknowledge support by NSF award CCF-2112665 (TILOS AI Institute) and NSF award 2134108.

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
