

Figure 2: **Trace of Hessian and Nuclear Norm.** Label noise SGD recovers the min nuclear norm solution via its sharpness-minimization implicit regularization and thus leads to better generalization (see Figure 1).

## A Experiments

In this section, we examine our theoretical results with controlled experiments via synthetic data. The experiments are based on mini-batch SGD and label noise SGD [6]. Both use the standard update rule $\boldsymbol{W}_{t+1} = \boldsymbol{W}_t - \eta\nabla\mathcal{L}_t(\boldsymbol{W}_t)$, but with different objectives:

- Mini-batch loss: $\mathcal{L}_t^{\text{mini-batch}}(\boldsymbol{W}) = \frac{1}{B}\sum_{i\in\mathcal{B}_t}(f_i(\boldsymbol{W}) - b_i)^2$;
- Label-noise loss: $\mathcal{L}_t^{\text{label-noise}}(\boldsymbol{W}) = \frac{1}{B}\sum_{i\in\mathcal{B}_t}(f_i(\boldsymbol{W}) - b_i + \xi_{t,i})^2$,

where $\mathcal{B}_t$ is the batch of size $B$ independently sampled with replacement at step $t$ and $\xi_t \in \mathbb{R}^d$ are i.i.d. multivariate zero-mean Gaussian random variables with unit variance.

It is known that with a small learning rate, label noise SGD implicitly minimizes the trace of Hessian of the loss, after reaching zero loss [10, 29]. In particular, Li et al. [29] show that after reaching zero loss, in the limit of step size going to zero, label noise SGD converges to a gradient flow according to the negative gradient of the trace of Hessian of the loss. As a result, we expect label noise SGD to be biased to regions with smaller trace of Hessian. We also compare the label noise SGD with vanilla SGD without label noise as a baseline, which can potentially find a solution with large sharpness when the learning rate is small. Note this is not contradictory with the common belief that mini-batch SGD prefers flat minimizers and thus benefits generalization [25, 23]. For example, assuming the convergence of mini-batch SGD, [50] shows that the solution found by SGD must have a small sharpness, bounded by a certain function of the learning rate. However, there is no guarantee when the learning rate is small and the upper bound of sharpness becomes vacuous.

In our synthetic experiments, we sample $n = 600$ input matrices $\{A_i\}_{i=1}^n$, where $A_i \in \mathbb{R}^{d\times d}$ with $d = 60$. Each entry $A_i^{(j,k)}$ is i.i.d. sampled from normal distribution $\mathcal{N}(0,1)$. The ground truth matrix $M^*$ is constructed by $M^* = M_1 M_2/d$, where $M_1 \in \mathbb{R}^{d\times r}$ and $M_2 \in \mathbb{R}^{r\times d}$ and $r$ is the rank of $M^*$. The entries in $M_1$ and $M_2$ are again i.i.d. sampled from $\mathcal{N}(0,1)$ and the rank $r$ is set to 3. The corresponding label is therefore computed via $b_i = \langle A_i, M^*\rangle$. The parameters $(W_1, ..., W_L)$ are sampled from a zero-mean normal distribution for depth $L = 2, 3, 5,$ and 10. For label noise SGD, we optimize the parameter via SGD with label noise drawn from $\mathcal{N}(0,1)$ and batch size 50. The learning rate is set to 0.01.

We examine our theory by plotting the training and testing loss along with the nuclear norm and the trace of Hessian of the label noise SGD solutions in Figures 1 and 2. As the figure illustrates, the trace of the Hessian exhibits a gradual decrement, eventually reaching a state of convergence over the course of the training process. This phenomenon co-occurs with the decreasing of the nuclear norm of the end-to-end matrix. In particular, we further plot the nuclear norm of the min nuclear norm solution obtained via solving convex optimization in Figure 2 and demonstrate that label noise

SGD converges to the minimal nuclear norm solution, as predicted by our theorem Theorem 1. As a consequence of this sharpness-minimization implicit bias, the test loss decreases drastically.

Interestingly, there are a few large spikes in the training loss curve of mini-batch SGD without label noise even after the training loss becomes as small as $10^{-12}$ and its generalization improves immediately after recovering from the spike. Meanwhile, the trace of hessian and the nuclear decrease during this process. We do not have a complete explanation for such spikes. One possible explanation from the literature [36] is that the loss landscape around the minimizers is too sharp and thus mini-batch SGD is not linear stable around the minimizer, so it escapes eventually. However, this explanation does not explain why minibatch SGD can find a flatter minimizer each time after escaping and re-converging.

## B Additional Related Work

**Implicit Bias of Gradient Descent on Matrix Factorization.** At first glance, overfitting could happen when the number of linear measurements is less than the size of the groundtruth matrix. Surprisingly, a recent line of works [20, 3, 19, 28, 40, 5, 22, 41] has shown that GD starting from small initialization has a good implicit bias towards solutions with approximate recovery of ground truth. Notably, Gunasekar et al. [20] show that for depth 2, GD from infinitesimal initialization is implicitly biased to the minimum nuclear norm solution under commuting measurements and Arora et al. [3] generalize this results to deep matrix factorization for any depth. This is very similar to our main result that for all depth ($\geq 2$) the implicit regularization is minimizing nuclear norm, though the settings are different. Moreover, when the measurements satisfy RIP, Li et al. [27], Stöger and Soltanolkotabi [45] show that GD exactly recovers the ground truth.

**Provable Generalization of Flatness Regularization for Two-layer Models.** To our best knowledge, most existing generalization analysis for flat regularization are for two-layer models, *e.g.*, Li et al. [29] shows that the min trace of hessian interpolating solution of 2-layer diagonal linear networks can recover sparse ground truth on gaussian or boolean data, and Nacson et al. [37] proves a generalization bound for the interpolating solutions with the smallest maximum eigenvalue of Hessian for non-centered data. Ding et al. [12] is probably the most related work to ours, which shows that the trace of Hessian implicit bias for two-layer matrix factorization is a rescaled version of the nuclear norm of the end-to-end matrix. Using this formula, they further prove that the flattest solution in this problem recovers the low-rank ground truth. However, matrix factorization with more than two layers is fundamentally more challenging compared to the depth two case; while we managed to obtain a formula for the trace of Hessian for deeper networks given a single measurement (see Theorem 7), as far as we know, one in general cannot obtain a closed-form solution for the trace of Hessian regularizer as a function of the end-to-end matrix for multiple measurements. In this work, we discover a way to bypass this hardness by showing that minimizing the trace of Hessian regularizer for a fixed end-to-end matrix approximately amounts to the nuclear norm of the end-to-end matrix, when the linear measurements satisfy the RIP property. As a cost of this approximation, we are not able to show the exact recovery of the low-rank ground truth, but only up to a certain precision.

**Sharpness Minimization in Deep Diagonal Linear Network.** Ding et al. [12] show that the minimizer of trace of Hessianin a deep diagonal matrix factorization model with Gaussian linear measurements becomes the Schatten $2 - 2/L$ norm of a rescaled version of the end to end matrix. At first glance, their result might seem contradictory to our result in the RIP setup, as their implicit regularization is not always the Nuclear norm — the sparsity regularization vanishes when $L \to \infty$. Similar results have been obtained by Nacson et al. [37] for minimizing a different notion of sharpness among all interpolating solutions, the largest eigenvalue of Hessian, on the same diagonal linear models. The subtle difference is that since we consider the more standard setting without assuming the weight matrices are all diagonal, then in the calculation of the trace of Hessian of the loss we need to also differentiate the loss with respect to the non-diagonal entries, even though their values are zero, which is quite different from $\ell_p$ norm regularization. This curiously shows the complicated interplay between the geometry of the loss landscape and the implicit bias of the algorithm.

**Sharpness-related Generalization Bounds.** Most existing sharpness-related generalizations depend on not only the sharpness of the training loss but also other complexity measures like a norm of the parameters or even undesirable dependence on the number of parameters [15, 46, 47, 17, 39].

In contrast, our result only involves the trace of Hessian but not parameter norm or the number of parameters, *e.g.*, our result holds for any (large) width of intermediate layers, $d_1, \ldots, d_{L-1}$.

## C  Proof of Lemma 3

*Proof.* For a fixed $j \in \{2, \ldots, L-1\}$ and vectors $x \in \mathbb{R}^{d_0}$ and $y \in \mathbb{R}^{d_L}$ we apply the RIP property in Definition 3 for the rank one matrix $X = xy^T$. As a result we get

$$(1-\delta)\|xy^T\|_F^2 \leq \frac{1}{n}\sum_{i=1}^n \langle A_i, xy^T \rangle^2 \leq (1+\delta)\|xy^T\|_F^2,$$

or equivalently

$$(1-\delta)\|x\|^2\|y\|^2 \leq \frac{1}{n}\sum_{i=1}^n (x^T A_i y)^2 \leq (1+\delta)\|x\|^2\|y\|^2. \tag{25}$$

Now for arbitrary indices $1 \leq \ell \leq d_{j-1}$ and $1 \leq k \leq d_j$, we pick $x, y$ in Equation (25) equal to the $\ell$th row of the matrix $W_{j-1} \ldots W_1$ and the $k$th column of the matrix $W_L \ldots W_{j+1}$:

$$(1-\delta)\|(W_{j-1} \ldots W_1)_{\ell:}\|^2\|(W_L \ldots W_{j+1})_{:k}\|^2$$

$$\leq \frac{1}{n}\sum_{i=1}^n ((W_{j-1} \ldots W_1)_{\ell:}A_i(W_L \ldots W_{j+1})_{:k})^2$$

$$\leq (1+\delta)\|(W_{j-1} \ldots W_1)_{\ell:}\|^2\|(W_L \ldots W_{j+1})_{:k}\|^2. \tag{26}$$

Summing this over all $\ell : k$, we obtain that the sum of Frobenius norm of matrices $W_{j-1} \ldots W_1 A_i W_L \ldots W_{j+1}$ concentrate around $\|W_{j-1} \ldots W_1\|_F^2\|W_L \ldots W_{j+1}\|_F^2$.

$$(1-\delta)\|W_{j-1} \ldots W_1\|_F^2\|W_L \ldots W_{j+1}\|_F^2$$

$$\leq \frac{1}{n}\sum_{i=1}^n \|W_{j-1} \ldots W_1 A_i W_L \ldots W_{j+1}\|_F^2$$

$$\leq (1+\delta)\|W_{j-1} \ldots W_1\|^2\|W_L \ldots W_{j+1}\|^2. \tag{27}$$

For $j = 1$, we apply Equation (25) with $x = (W_{j-1} \ldots W_1)_{\ell:}$ and $y = e_k$, where $e_k$ is the $k$th standard vector:

$$(1-\delta)\|(W_{L-1} \ldots W_1)_{\ell:}\|^2 \leq \frac{1}{n}\sum_{i=1}^n ((W_{L-1} \ldots W_1)_{\ell:}A_i e_k)^2 \leq (1+\delta)\|(W_{L-1} \ldots W_1)_{\ell:}\|^2.$$

Summing this for all $k, \ell$

$$(1-\delta)d_0\|W_{L-1} \ldots W_1\|_F^2 \leq \frac{1}{n}\sum_{i=1}^n \|W_{L-1} \ldots W_1 A_i\|_F^2 \leq (1+\delta)d_0\|W_{L-1} \ldots W_1\|_F^2. \tag{28}$$

Similarly for $j = L$

$$(1-\delta)d_L\|W_L \ldots W_2\|_F^2 \leq \frac{1}{n}\sum_{i=1}^n \|A_i W_L \ldots W_2\|_F^2 \leq (1+\delta)d_L\|W_L \ldots W_2\|_F^2. \tag{29}$$

Combining Equations (27), (28), and (29)

$$(1-\delta)R(W) \leq \text{tr}(\nabla^2 L)(W) \leq (1+\delta)R(W).$$

$\square$

# D   Proof of Theorem 7

*Proof of Theorem 7.*   Recall that we hope to characterize the solution with a minimal trace of hessian given that the end-to-end matrix $E(\boldsymbol{W}) = W_L \cdots W_1$ is equal to some fixed matrix $M$, namely,

$$\min_{E(\boldsymbol{W})=M} \sum_{i=1}^{L} \|W_{i-1} \ldots W_1 A^T W_L \ldots W_{i+1}\|_F^2.$$

Let $\boldsymbol{W}$ be any minimizer of the above objective. For arbitrary matrix $C \in \mathbb{R}^{d_i \times d_i}$, define

$$U(t) = \exp(tC) \triangleq \sum_{i=0}^{\infty} \frac{(tC)^i}{i!},$$

For any $i$, we multiply $W_i$ from left by $U(t)$ and multiply $W_{i+1}$ by $U(t)^{-1}$ from right,

$$W_i(t) \leftarrow U(t)W_i,$$
$$W_{i+1}(t) \leftarrow W_{i+1}U(t)^{-1}.$$

For convenience, below we drop the dependence of $W_i(t), W_{i+1}(t)$ over $t$, that is, only $W_i$ and $W_{i+1}$ are implicitly functions of $t$, while the rest $W_j$ are independent of $t$. Then, note that for any $j \leq i-1$ we have

$$W_{j-1} \ldots W_1 A^T W_L \ldots W_{i+1}U(t)^{-1}U(t)W_i \ldots W_{j+1} = W_{j-1} \ldots W_1 A^T W_L \ldots W_{j+1},$$

and for $j \geq i+2$:

$$W_{j-1} \ldots W_{i+1}U(t)^{-1}U(t)W_i \ldots W_1 A^T W_L \ldots W_{j+1} = W_{j-1} \ldots W_{i+1}W_i \ldots W_1 A^T W_L \ldots W_{j+1}.$$

So the only terms that actually change as a function of $t$ correspond to $j = i$,

$$\|W_{i-1} \ldots W_1 A^T W_L \ldots W_{i+1}\|_F^2 = \mathrm{tr}(W_{i-1} \ldots W_1 A^T W_L \ldots W_{i+1}W_{i+1}^T \ldots W_L^T A W_1^T \ldots W_{i-1}^T), \tag{30}$$

and to $j = i+1$,

$$\|W_i \ldots W_1 A^T W_L \ldots W_{i+2}\|_F^2 = \mathrm{tr}(W_i \ldots W_1 A^T W_L \ldots W_{i+2}W_{i+2}^T \ldots W_L^T A W_1^T \ldots W_i^T). \tag{31}$$

Now taking derivative of $U(t)$ with respect to $t$,

$$U'(0) = C.$$

Now for every $j \in \{1, \ldots, L\}$ we define

$$\widetilde{W}_j = W_{j-1} \ldots W_1 A^T W_L \ldots W_{j+1},$$

where we use $W_{i-1} \ldots W_1$ and $W_L \ldots W_{i+1}$ to denote identity for $i = 1$ and $i = L$ respectively.

Then, if we take derivative from the terms (30) and (31) with respect to $t$:

$$\frac{d}{dt}\|W_{i-1} \ldots W_1 A^T W_L \ldots W_{i+1}\|_F^2 \Big|_{t=0}$$
$$= -\mathrm{tr}((C + C^T)W_{i+1}^T \ldots W_L^T A W_1^T \ldots W_{i-1}^T W_{i-1} \ldots W_1 A^T W_L \ldots W_{i+1}), \tag{32}$$
$$= \mathrm{tr}((C + C^T)\widetilde{W}_i^T \widetilde{W}_i).$$

and

$$\frac{d}{dt}\|W_i \ldots W_1 A^T W_L \ldots W_{i+2}\|_F^2 \Big|_{t=0}$$
$$= -\mathrm{tr}((C + C^T)\widetilde{W}_{i+1}\widetilde{W}_{i+1}^T) \tag{33}$$

Now from the optimality of $\boldsymbol{W}$, the following equality holds for every matrix $C \in \mathbb{R}^{d \times d}$:

$$\frac{d}{dt}\mathrm{tr}[\nabla^2 \mathcal{L}(\boldsymbol{W}(t))]\Big|_{t=0} = \mathrm{tr}((C + C^T)(\widetilde{W}_i^T \widetilde{W}_i - \widetilde{W}_{i+1}\widetilde{W}_{i+1}^T)) = 0. \tag{34}$$

Now since $C$ is arbitrary and the matrices $\widetilde{W}_i^T \widetilde{W}_i$ and $\widetilde{W}_{i+1} \widetilde{W}_{i+1}^T$ are symmetric, we must have

$$\widetilde{W}_i^T \widetilde{W}_i = \widetilde{W}_{i+1} \widetilde{W}_{i+1}^T. \tag{35}$$

Equation (35) implies that all $\widetilde{W}_i$ for $1 \le i \le L$ have the same set of singular values. Moreover, there exists matrices $\{U_i\}_{i=0}^L$ where the columns of each matrix are orthogonal, such that for each $1 \le i \le L$,

$$\widetilde{W}_i = W_{i-1} \dots W_1 A^T W_L \dots W_{i+1} = U_{i-1} \Lambda U_i^T. \tag{36}$$

Multiplying Equation (35) for all $1 \le i \le L$ (in the case $i = 1$ we take $W_1 \dots W_{i-1}$ as identity), we get

$$\left( A^T E(\mathbf{W}) \right)^{L-1} A^T = \left( A^T W_L \dots W_1 \right)^{L-1} A^T = U_0 \Lambda^L U_L^T, \tag{37}$$

or in case where $A$ is positive semi-definite,

$$A^{1/2} \left( A^{1/2} E(\mathbf{W}) A^{1/2} \right)^{L-1} A^{1/2} = U_0 \Lambda^L U_L^T. \tag{38}$$

But having access to Equations (36), we can write $\mathrm{tr}[\nabla^2 \mathcal{L}(\boldsymbol{W})]$ at the minimizer point $\boldsymbol{W} = (W_1, \dots, W_L)$ as

$$\sum_{i=1}^L \| W_{i-1} \dots W_1 A W_L \dots W_{i+1} \|_F^2 = L \|\Lambda\|_F^2 = L \|\Lambda^L\|_{S_{2/L}}^{2/L}.$$

which based on Equation (37) is equal to

$$L \left\| \left( A^T E(\mathbf{W}) \right)^{L-1} A^T \right\|_{S_{2/L}}^{2/L},$$

or in the symmetric case is equal to

$$L \left\| A^{1/2} \left( A^{1/2} E(\mathbf{W}) A^{1/2} \right)^{L-1} A^{1/2} \right\|_{S_{2/L}}^{2/L}.$$

This is the induced regularizer of the trace of Hessian over all interpolating solutions for linear network with depth $L$ in the space of end-to-end matrices.

$\square$

## E   Other Omitted Proofs

### E.1   Proof of Theorem 3

*Proof.* By Corollary 1, we know that with probability at least $1 - \exp(\Omega(n))$,

$$\|E(\boldsymbol{W}^*)\|_* \le 3\|M^*\|_*.$$

Note by assumption, $n = \Omega(d_0 + d_L)$. Thus it suffices to show that with probability at least $1 - \exp(\Omega(d_0 + d_L))$, for all interpolating solutions in $\mathcal{H}_{3\|M^*\|_*} = \{h_M \mid \|M\|_* \le 3 \|M^*\|_*\}$, Equation (23) holds.

Recall $\bar{\mathcal{L}}\big(E(\boldsymbol{W})\big)$ is the population squared loss of the end-to-end matrix $E(\boldsymbol{W}) \in \mathbb{R}^{d_0 \times d_L}$. Namely,

$$\bar{\mathcal{L}}(E(\boldsymbol{W})) \triangleq \mathbb{E}_A (\langle A, E(\boldsymbol{W}^*)\rangle - \langle A, M^*\rangle)^2 = \mathbb{E}_A \langle A, E(\boldsymbol{W}) - M^* \rangle^2 = \|E(\boldsymbol{W}^*) - M^*\|_F^2.$$

First, we bound the population Rademacher complexity of $\mathcal{H}_{3\|M^*\|_*}$. The empirical Rademacher complexity on $\{A_i\}_{i=1}^n$ is

$$\mathcal{R}_n(\mathcal{H}_{3\|M^*\|_*}) = \frac{1}{n} \mathbb{E}_{\epsilon \sim \{\pm 1\}^n} \sup_{h \in \mathcal{H}_{3\|M^*\|_*}} \sum_{i=1}^n \epsilon_i h(A_i)$$

$$= \frac{1}{n} \mathbb{E}_{\epsilon \sim \{\pm 1\}^n} \sup_{M: \|M\|_* \le 3\|M^*\|_*} \sum_{i=1}^n \langle \epsilon_i A_i, M \rangle = 3/n \cdot \|M^*\|_* \left\| \sum_{i=1}^n \epsilon_i A_i \right\|_2.$$

Note that the matrix $A_{sum} = \sum_{i=1}^{n} \epsilon_i A_i$ itself is an iid Gaussian matrix where each entry is sampled from $\mathcal{N}(0, n)$. Hence, from Proposition 2.4 in [43], we have the following tail bound on the spectral norm of $A_{sum}$

$$\mathbb{P}(\|A_{sum}\|_2 \geq c_1 \sqrt{n}(\sqrt{d_0} + \sqrt{d_L}) + \sqrt{n}t) \leq 2e^{-c_2 t^2}, \tag{39}$$

This implies $\mathbb{E}\|A_{sum}\|_2 = O\big(\sqrt{n}(\sqrt{d_0} + \sqrt{d_L})\big)$, which in turn bounds the Rademacher complexity

$$\overline{\mathcal{R}}_n(\mathcal{H}_{3\|M^*\|_*}) = \mathbb{E}\,\mathcal{R}_n(\mathcal{H}_{3\|M^*\|_*}) = O\left(\frac{\sqrt{d_0} + \sqrt{d_L}}{\sqrt{n}}\|M^*\|_*\right). \tag{40}$$

Note that the Gaussian distribution $\mathcal{G}_{d_L \times d_0}$ is unbounded, which makes the value of the squared loss unbounded, while it is convenient to bound the generalization gap when the value of the loss is bounded. To cope with this fact, for a given threshold $c$, we define a truncated version of the loss denoted by $l_c(x, y) = \ell_c(x - y)$, plotted in Figure 3, which is a smooth approximation of the squared loss.

$$l_c(x, y) = \ell_c(x - y) = \begin{cases} (x - y)^2, & \text{if } x - y \in [-c, c], \\ -(x - y)^2 + 4c|x - y| - 2c^2, & \text{if } x - y \in [-2c, -c] \cup [c, 2c], \\ 2c^2, & \text{if } x - y \in (-\infty, -2c] \cup [2c, \infty). \end{cases} \tag{41}$$

It is easy to verify $\partial_x \ell_c$ is 2-lipschitz in $x$. Also it is clear that $l_c(x, y) \leq \max(2c^2, l(x, y))$ for all $x, y$ and $l_c(x, y) < l(x, y)$ only when $|x - y| > c$. Next, we define the $c$-cap population loss $\bar{\mathcal{L}}_c$ with respect to $\ell_c$:

$$\bar{\mathcal{L}}_c(M) = \mathbb{E}_{A \sim \mathcal{G}_{d_L \times d_0}} \ell_c(\langle A, M \rangle, \langle A, M^* \rangle).$$

Thus we have

$$\bar{\mathcal{L}}(M) - \bar{\mathcal{L}}_c(M) \leq \mathbb{E}\mathbb{1}_{|\langle A, M - M^* \rangle| \geq c} \langle A, M - M^* \rangle^2.$$

But note that the variable $\langle A, M - M^* \rangle$ is a Gaussian variable with variance $\|M - M^*\|_F^2 \leq \|M - M^*\|_*^2$. Hence, from Lemma 6, picking $c = \Theta(\log(n)\|M^*\|_*)$, for all $M \in \mathcal{H}_{3\|M^*\|_*}$,

$$0 \leq \bar{\mathcal{L}}(M) - \bar{\mathcal{L}}_c(M) \leq O(\frac{\|M^*\|_*^2 \log n}{n}).$$

Now using the Rademacher complexity bound in (40) and applying Theorem 6, we have for all interpolating solutions $M \in \mathcal{H}_{3\|M^*\|_*}$ with probability at least $1 - \exp(\Omega(d_0 + d_L))$:

$$\bar{\mathcal{L}}_c(M) \leq O\left(H \log^3(n)\mathcal{R}_n^2 + \frac{c^2(d_0 + d_L)}{n}\right)$$

$$\leq O\left(\|M^*\|_*^2 \frac{(d_0 + d_L) \log^3 n}{n}\right) \tag{42}$$

where $H$ is the gradient smoothness of the loss $\ell_c$ which is 2 and $\mathcal{L}_c$ is the empirical loss defined in (2) with square loss substituted by $\ell_c$. Above, we used the fact that $\ell_c$ is bounded by $2c^2$. $\qquad\square$

**Lemma 6.** *For standard Gaussian variable $X$, we have*

$$\mathbb{E}\mathbb{1}_{|X| \geq c} X^2 \leq e^{-c^2/2} \frac{2(c^2 + 2)}{c\sqrt{2\pi}}.$$

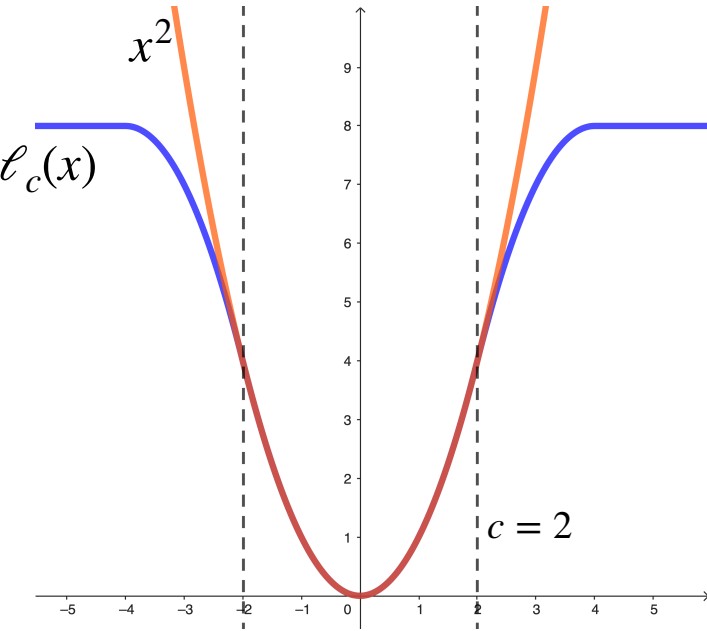

Figure 3: The smooth surrogate loss $\ell_c$ as defined in Equation (41) with parameter $c = 2$.

*Proof of Lemma 6.*

$$
\begin{aligned}
\mathbb{E}\mathbb{1}_{|X|\geq c}X^2 &= 2/\sqrt{2\pi}\int_{x=c}^{\infty} x^2 e^{-\frac{x^2}{2}}\,dx \\
&\leq 2/\sqrt{2\pi}\int_{x=c}^{\infty} \frac{x^3}{c} e^{-\frac{x^2}{2}}\,dx \\
&= 1/(c\sqrt{2\pi})\int_{x=c}^{\infty} x^2 e^{-\frac{x^2}{2}}\,dx^2 \\
&= 1/(c\sqrt{2\pi})\int_{x=c^2}^{\infty} x e^{-\frac{x}{2}}\,dx \\
&= 1/(c\sqrt{2\pi})(-0 - (-2e^{-c^2/2}(c^2+2))) \\
&= e^{-c^2/2}\frac{2(c^2+2)}{c\sqrt{2\pi}}.
\end{aligned}
$$

$\square$

### E.2 Proof of Lemma 5

*Proof of Lemma 5.* Consider its SVD decomposition of $M$, $M = \sum_{i=1}^{d} \alpha_i u_i v_i^T$, where $\alpha_i$'s are the singular values and $\{u_i\}_{i=1}^d$, $\{v_i\}_{i=1}^d$ each is an orthonormal basis for $\mathbb{R}^d$. We can write

$$
\begin{aligned}
\sum_{i=1}^{n}\langle A_i, M\rangle^2 &= \frac{1}{n}\sum_{i=1}^{n}(\sum_{j=1}^{d}\alpha_j u_j^T A_i v_j)^2 \\
&= \frac{1}{n}\sum_{i=1}^{n}\sum_{j:k=1}^{d}\alpha_j\alpha_k \mathrm{tr}(A_i v_j u_j^T)\mathrm{tr}(A_i v_k u_k^T) \\
&= \sum_{j:k=1}^{d}\frac{1}{4n}\sum_{i=1}^{n}\alpha_j\alpha_k\big(\mathrm{tr}(A_i(v_j u_j^T + v_k u_k^T))^2 - \mathrm{tr}(A_i(v_j u_j^T - v_k u_k^T))^2\big).
\end{aligned}
$$

But again using the $(2, \delta)$-RIP of $\{A_i\}_{i=1}^n$,

$$(1 - \delta)\|v_j u_j^T + v_k u_k^T\|_F^2 \leq \frac{1}{4n}\sum_{i=1}^n \mathrm{tr}(A_i(v_j u_j^T + v_k u_k^T))^2 \leq (1 + \delta)\|v_j u_j^T + v_k u_k^T\|_F^2$$

$$(1 - \delta)\|v_j u_j^T - v_k u_k^T\|_F^2 \leq \frac{1}{4n}\sum_{i=1}^n \mathrm{tr}(A_i(v_j u_j^T - v_k u_k^T))^2 \leq (1 + \delta)\|v_j u_j^T - v_k u_k^T\|_F^2.$$

This implies

$$\frac{1}{4n}\sum_{i=1}^n \left(\mathrm{tr}(A_i(v_j u_j^T + v_k u_k^T))^2 - \mathrm{tr}(A_i(v_j u_j^T - v_k u_k^T))^2\right)$$

$$\leq \frac{1}{2}\delta(\|v_j u_j^T\|_F + \|v_k u_k^T\|_F) + (1 + \delta)\langle v_j u_j^T, v_k u_k^T\rangle.$$

Summing this from $j$ to $k$ and noting that $\langle v_j u_j^T, v_k u_k^T\rangle$ is zero for $j \neq k$:

$$\sum_{i=1}^n \langle A_i, M\rangle^2 \leq (1 + \delta)(\sum_{j=1}^d \alpha_j^2) + \delta(\sum_j |\alpha_j|)^2 \leq (1 + \delta)\|M\|_F^2 + \delta\|M\|_*^2. \qquad (43)$$

Similarly we obtain

$$\sum_{i=1}^n \langle A_i, M\rangle^2 \geq (1 - \delta)\|M\|_F^2 - \delta\|M\|_*^2. \qquad (44)$$

Combining Equations (43) and (44):

$$\left|\sum_{i=1}^n \langle A_i, M\rangle^2 - \|M\|_F^2\right| \leq \delta\|M\|_F^2 + \delta\|M\|_*^2 \leq 2\delta\|M\|_*^2. \qquad (45)$$

This completes the proof. $\qquad\qquad\qquad\qquad\qquad\qquad\qquad\qquad\qquad\qquad\qquad\qquad\qquad$ $\square$

# F   Proof of Theorem 4

*Proof of Theorem 4.* Here we view matrices in $\mathbb{R}^{d_0 \times d_L}$ as $d_0 d_L$ dimensional vectors, hence by rotating a matrix with an orthogonal transformation we mean to rotate the corresponding vector. Note that the minimum $\ell_2$ solution of the regression problem is given by $\widetilde{M}$ defined as

$$\widetilde{M} = \sum_{i=1}^n A_i \left[\left(\left(\langle A_i, A_j\rangle\right)_{1 \leq i,j \leq n}^{-1} b\right]_i.$$

First, note that if we rotate the ground-truth matrix $M^*$ with an arbitrary orthogonal matrix $U$, then $\widetilde{M}$ rotates according to the same $U$. Combining this with the fact the distribution on the measurement matrices is Gaussian and rotationally symmetric, we conclude that the population loss $\mathcal{L}'(\widetilde{M})$ is the same for all $M^*$. Hence, to lower bound the population loss, we can further assume that the entries of $M^*$ are sampled from standard Gaussian distribution. Hence, for any $M^*$ we can write

$$\mathbb{E}_{\{A_i\}_{i=1}^n}\mathcal{L}'(\widetilde{M}) = \mathbb{E}_{M^*}\mathbb{E}_{\{A_i\}_{i=1}^n}\mathcal{L}'(\widetilde{M})$$

$$= \mathbb{E}_{\{A_i\}_{i=1}^n}\mathbb{E}_{M^*}\|\widetilde{M} - M^*\|_F^2$$

$$= \mathbb{E}_{\{A_i\}_{i=1}^n}(1 - \frac{n}{d_0 d_L})\|M^*\|_F^2$$

$$= (1 - \frac{n}{d_0 d_L})\|M^*\|_F^2.$$

where we used the fact that $\widetilde{M}$ is the projection of $M^*$ onto the subspace spanned by $\{A_i\}_{i=1}^n$. $\quad\square$