# OpenReview forum: "What is the Inductive Bias of Flatness Regularization? A Study of Deep Matrix Factorization Models"
_NeurIPS.cc/2023/Conference — NeurIPS 2023 poster_

### Official Review · Reviewer_JwkB · 2023-06-15

**Soundness:** 4 excellent
**Presentation:** 3 good
**Contribution:** 3 good
**Rating:** 7
**Confidence:** 4

**Summary:**

The paper studies the minimizers of the loss surface of deep matrix factorization that have a minimal trace of the Hessian. The trace of the Hessian is a measure of flatness of the minimum, that is favored by SGD.

The authors show that for matrix sensing with observations that satisfy RIP (which is in particular true for enough Gaussian observations) the parameters that are the flatest have end-to-end matrix that approximately minimize the nuclear norm amongst fitting matrices. This in turn implies generalization guarantees that are significantly faster than the if one were to minimize the Frobenius norm instead of the nuclear norm.

**Strengths:**

The description of the flatness bias given by this paper is to my knowledge new, and some of the results are a bit unexpected.

The writing is clear and thorough and the proofs are easy to understand.

**Weaknesses:**

The results hangs on the fact that the trace of the Hessian is the right notion of flatness for SGD, which is to my knowledge not decided yet. For GD the relevant measure of flatness is instead the largest eigenvalue of the Hessian, which will probably lead to a quite different implicit bias. And since SGD can in some cases be quite similar to GD, it seems unlikely that the bias would simply be described by the trace of Hessian, instead this exact bias could change depending on the learning rate and batch size.

The lack of empirical experiments to test whether the bias described here is observed for SGD, label noise SGD or 1-SAM would help motivate this assumption.

**Questions:**

A shallow linear network with L2 regularization is also known to recover the minimal nuclear norm solution [Dai et al., Representation Costs of Linear Neural Networks: Analysis and Design]. Do you think that there is any advantage to rely on the flatness bias instead of the L2 regularization?

**Limitations:**

The biggest limitation is clearly the RIP condition, but it is well discussed.

---

> ### Author Rebuttal · Authors · 2023-08-09
>
> **implicit bias**: We would like to point out that in the paper [28] authors mathematically show that in the limit of step size going to zero, label noise SGD evolves according to a gradient flow according to the trace of hessian of the loss. The same fact can be seen for 1-SAM,  but the reviewer is right that it might not be the right notion for vanilla SGD.
>
> **Experiments**: In the final version, we will include additional experiments that show how the trace of Hessian evolves for SGD with and without label noise, see also the pdf included with this rebuttal. Indeed, we do observe that label noise SGD decreases the trace of the Hessian.

---

> > ### Comment · Reviewer_JwkB · 2023-08-10
> >
> > Thank you for the answer and additional experiments.

---

### Official Review · Reviewer_ecS4 · 2023-07-03

**Soundness:** 4 excellent
**Presentation:** 4 excellent
**Contribution:** 3 good
**Rating:** 7
**Confidence:** 4

**Summary:**

In the context of matrix sensing with deep matrix factorizations, the paper analyzes the inductive bias of interpolators with minimal Hessian trace, which is a well-known measure of sharpness. Specifically, under a Restricted Isometry Property (RIP) assumption on the linear measurements, it establishes that the minimal Hessian trace with which the deep matrix factorization can express an interpolator is approximately equal to the latter’s nuclear norm. In turn, this yields guarantees on the recovery of the ground truth matrix, as well as a separation result from the recovery obtained by the minimal Frobenius norm interpolator.

As an additional contribution, develops a closed form expression for the minimal Hessian trace of an interpolator when there is only a single linear measurement.

**Strengths:**

1. Well-written and easy to follow. The motivation, problem at hand, and main results are clearly described.

2. Helpful discussions are provided for each of the technical results, in particular regarding their implications and relation to existing work.

3. The technical contributions establish a new connection between flatness, in terms of minimal Hessian trace, and generalization for deep matrix factorizations (previous work of Ding et al. 2022 studies only depth two factorizations). Despite conventional wisdom and empirical evidence suggesting that flatness may lead to good generalization, formally showing that this is the case has proven challenging. This attests to the significance of the current paper’s results.
Personally, I found it interesting that depth does not help in the analyzed setting, in the sense that minimizing the Hessian trace amounts to approximately minimizing the nuclear norm, similar to the depth L = 2 case. This may suggest that deep matrix factorization with RIP measurements is an unsatisfactory setting for uncovering the benefits of depth in deep learning and its (possible) relation with flatness.


**Weaknesses:**

The points below pertain to the scope of the technical contributions, suggesting places for improvement. I do not believe that points (2) and (3) significantly harm the quality of the current paper and are perhaps considerations for future work. For (1), it seems that it may be straightforward to incorporate, in which case I believe it can increase the generality of the current work.

1. Only interpolators with exactly minimal Hessian trace are considered. Since in practice a more realistic view is that it is only approximately minimized, if possible, it is worth (even if in an appendix) extending the generalization results to interpolators whose Hessian trace is approximately minimal.

2. The current paper characterizes the regularizer induced by minimizing the Hessian trace only for interpolators. Since using explicit regularization can lead to solutions that don't entirely minimize the loss to zero, I believe it is important to understand the inductive bias of minimizing the Hessian trace for non-interpolating linear mappings as well. While more complex, it may be more realistic.

3. The current paper only treats the question of what interpolating the data with minimal Hessian trace implies, and not the complementary question of whether it, or other measures of sharpness more generally, are implicitly minimized in deep matrix factorization by standard optimizers.

Additional (minor) comments and typos:
- Typo in line 75: “regularzier” should be “regularizer”.
- Typo in line 101: I believe “observe” should have been “observed”.
- Typo in equation after line 166: I believe there is a missing gradient symbol in the middle expression and a factor of two in both the middle and right expressions.



**Questions:**

Have you considered other notions of sharpness besides the trace of the Hessian, e.g. its the maximal eigenvalue? In particular, can minimizing the maximal eigenvalue of the Hessian ensure generalization in the considered setting, or is it too weak and one must look at the trace of the Hessian?

**Limitations:**

The authors have adequately addressed limitations of the work.

---

> ### Author Rebuttal · Authors · 2023-08-09
>
> 1. **If possible, it is worth (even if in an appendix) extending the generalization results to interpolators whose Hessian trace is approximately minimal.**
>
>    **Reply**: We will include this generalization in the appendix in the final version of the work.
>
> 2. **Since using explicit regularization can lead to solutions that don't entirely minimize the loss to zero, I believe it is important to understand the inductive bias of minimizing the Hessian trace for non-interpolating linear mappings as well. While more complex, it may be more realistic.**
>
>    **Reply**: Note that for points where the loss is non-zero, the form of trace of hessian of the loss is not given by Lemma 2 and Equation (7). Therefore, our approach in approximating the minimizers of these terms given a fixed end-to-end matrix and recovering the nuclear norm of the end to end matrix is not valid anymore; namely, this approximation will further depend on the distance to the manifold of zero loss. While beyond the scope of this work, our guess is that given some measure of closeness to the manifold (i.e. close to satisfying the linear constraints), one should still be able to use the same approach to approximate the trace of hessian regularizer.
>
>
> 3. **The current paper only treats the question of what interpolating the data with minimal hessian trace implies, and not the complementary question of whether it, or other measures of sharpness more generally, are implicitly minimized in deep matrix factorization by standard optimizers.**
>
>     **Reply**: The problem of convergence of label noise SGD to the minimizer of trace of hessian for deep linear models is open and can be an interesting future direction.
>
>
>  **Typos**: Thanks for pointing them out, we will fix them in the final version.
>
> ## Question:
>
> **Have you considered other notions of sharpness besides the trace of the hessian, e.g. its maximal eigenvalue?**
>
>  **Reply**: The maximum eigenvalue regularizer can be obtained via different algorithms and a possible analysis for deep matrix factorization with maximum eigenvalue does not seem to be relevant to the analysis of the trace of hessian we are presenting, but is certainly an interesting future direction. Empirically, label noise SGD is observed to decrease the trace of hessian; this behavior is also proved mathematically in the limit of step size going to zero, in [28].

---

> > ### Comment · Reviewer_ecS4 · 2023-08-11
> >
> > Thank you for the response, I read it and the other reviews carefully.
> >
> > Accordingly, I would like to keep my initial positive assessment of the paper.

---

### Official Review · Reviewer_PGyv · 2023-07-06

**Soundness:** 3 good
**Presentation:** 3 good
**Contribution:** 3 good
**Rating:** 6
**Confidence:** 3

**Summary:**

The manuscript seeks to understand Hessian trace regularization in the case of deep linear network training with the mean squared error of linear measurements. It obtains a description of the effective regulariser which can be approximated and made more explicit in some cases. The manuscript obtains results on matrix recovery and generalization.

**Strengths:**

* The problem under investigation, to characterize solutions with minimum Hessian trace, is interesting.
* The work obtains results indicating that the number of measurements needed to obtain a good approximation of a target matrix is smaller when using trace regularization as opposed to l2 regularization of the end to end matrix.

**Weaknesses:**

* The presentation of the results does not make the assumptions sufficiently clear in a timely manner.
* The settings are restrictive. The networks are assumed to have no bottlenecks, so that the set of representable end to end matrices have no constraints. The considered objective function is a sum of squared errors of linear measurements, whereby it is assumed that the measurements satisfy an RIP property, or that the network has only two layers, or that there is only one measurement.
* The theoretical results could have been strengthened by numerical experiments, particularly given the restricted conditions for which the theoretical results are obtained.

**Questions:**

* In Theorem 1 it is indicated that the data should satisfy an RIP assumption, but this condition is not described explicitly. The (1,\delta)-RIP condition should be described explicitly before or at the time of stating the theorem.
* After Theorem 1 it is stated that in more general cases it is challenging to compute F, but it is not clearly stated what is meant by more general cases.
* In Theorem 2, what is the dependence on n?
* In Theorem 3, what is r? This seems to be introduced 3 pages later in Definition 3.
* Theorem 4 appears to consider regularization of the end to end matrix. What would be the result if one instead minimises the l2 norm of the parameters (factor matrices)?
* There are typos in the display equation following line 166.
* In Theorem 5 the argument of the Hessian of the Loss should be the factor matrices?
* Example 1, missing negative sign in exponent?
* Please add an explanation for the second inequality in (13).
* In proof of Theorem 1, argument is W_1,\ldots, W_L instead of M?

**Limitations:**

* The work considers deep linear networks with no bottlenecks, in which case the image function space contains all matrices.
* It would be interesting to combine and compare the discussion of trace of Hessian minimisation with l2 regularization of the parameters.

---

> ### Author Rebuttal · Authors · 2023-08-09
>
> 1. **The presentation of the results does not make the assumptions sufficiently clear in a timely manner.**
>
>      **Reply**: We would like to point out that the RIP and width assumptions are mentioned in pages 2 and 3, but we are happy to follow the reviewers’ suggestions on stating them earlier.
>
> 2. **The settings are restrictive.**
>
>      **Reply**: In the general case of multiple measurements, that we analyze here, the trace of hessian regularizer does not have a closed form, so it is very challenging to analyze (e.g., prior works make strong assumptions such as 2 layers). Yet, we manage to approximate it with a function of the nuclear norm under RIP, which is highly non-trivial. Of course, it is an interesting future direction to understand the behavior of the implicit bias in other remaining settings, e.g. when the distribution of the matrices are heavy-tailed.
>
> 3. **The theoretical results could have been strengthened by numerical experiments**:
>
>      **Reply**: Some preliminary experiments are included in Appendix F. We will include more detailed experiments in the final version, see the plots in the included pdf.
>
> ## Questions:
> 1. **The (1,$\delta$)-RIP condition should be described explicitly before or at the time of stating the theorem.**
>
>    **Reply**: We will define RIP before the main result to make it clearer for the reader.
>
> 2. **After Theorem 1 it is stated that in more general cases it is challenging to compute F, but it is not clearly stated what is meant by more general cases.**
>
>      **Reply**: By more general cases, we mean beyond the one-measurement case and the RIP condition for multiple-measurements. As we mentioned, the case of multiple measurements for depth more than two (unlike the depth two case) is not solvable in closed form. We will clarify this in the final version.
>
> 3. **In Theorem 2, what is the dependence on n?**
>
>      **Reply**: Note that in general in the definition of RIP (Definition 3) $\delta$ implicitly depends on $n$ and this dependency can vary for different distributions. But for example, for the Gaussian case we have $\delta(n) = \sqrt{\frac{d_L + d_0}{n}}$ as stated right after Theorem 2.
>
> 4. **In Theorem 3, what is r? This seems to be introduced 3 pages later in Definition 3.**
>
>      **Reply**: $r$ should be set to one here, thanks for pointing to this typo.
>
> 5. **Theorem 4 appears to consider regularization of the end to end matrix. What would be the result if one instead minimizes the l2 norm of the parameters (factor matrices)?**
>
>      **Reply**: While in this work we are interested to see the effect of trace of hessian regularization as a function of the end to end matrix, the problem of regularizing the factor matrices, e.g., with l2 as the reviewer mentioned, is indeed interesting on its own. We conjecture the induced regularizer of $\ell_2$ norm minimization is the Schatten-$2/L$- (semi)norm of the end-to-end matrix, though we don’t have a proof for depth $L$ larger than $3$.
>
>
> 6. **There are typos in the display equation following line 166.**
>
>    **Reply**: Thanks for pointing this out, we will add the missing gradient to this equation.
>
> 7. **In Theorem 5 the argument of the Hessian of the Loss should be the factor matrices?**
>
>    **Reply**:We see $\min\{W_1W_2=M\}tr[\nabla^2 \mathcal L]$ in total as a function of $M$, which is why we put its argument to be $M$.
>
> 8. **Example 1, missing negative sign in exponent?**
>
>     **Reply**: Thanks for pointing this out, we will fix it in the final version.
>
> 9. **Please add an explanation for the second inequality in (13).**
>
>    **Reply**: That is a rearrangement of the terms and indeed it is an equality, we will clarify it further in the final version.
>
> 10. **In proof of Theorem 1, argument is $W_1,\ldots, W_L$ instead of $M$?**
>
>      **Reply**: Same response as case 7 above.

---

> > ### Comment · Reviewer_PGyv · 2023-08-20
> > **Response to rebuttal**
> >
> > I appreciate the authors response to my initial review, particularly about organisation of definitions, a few derivations, and typos. My general assessment of the manuscript remains unchanged, leaning accept.

---

### Official Review · Reviewer_sgkx · 2023-07-16

**Soundness:** 3 good
**Presentation:** 4 excellent
**Contribution:** 2 fair
**Rating:** 4
**Confidence:** 2

**Summary:**

This work considers the implicit regularization of deep neural networks with linear activations and linear data. While previous works have shown that stochasticity in optimizers has the effect of smoothing the loss function, this work derives how this less sharp loss function can result in better generalization performance under certain assumptions. Specifically, the authors derive the induced regularizer in three cases: (1) two-layer linear networks, (2) networks learned with a single example, and (3) data that satisfies the restricted isometry property. Then, in these settings, generalization bounds are derived.

**Strengths:**

* I found this work to be very well presented and clear. In general, I found the presentation made the content quite accessible for a reader not familiar with the literature. I appreciate the author's effort to ensure the content was as self-contained as possible with all required background knowledge neatly included.
* I found Sec. 1.1 summarizing the main results of the work to be extremely helpful before proceeding into the details later.
* Notation was well-considered, clear, and consistent.
* The related work section was extensive between the main text and the appendix. Although I am not familiar with this literature, this section provided a helpful summary (although I cannot comment on its accuracy).
* The claims made in the abstract are an accurate reflection of the paper's contributions.
* Some synthetic numerical experiments are provided to validate the theoretical findings.

**Weaknesses:**

* My single major issue with this work is with respect to its practical significance. Almost the entire contribution of this work lies in the theoretical results derived within. However, all of these results are derived under extremely restrictive assumptions/conditions. My understanding is that they assume (at a minimum):
   - A neural network with linear activations.
   - The regression setting (I believe results would not hold under e.g. cross-entropy loss?)
   - The relationship between the observations and their ground truth targets is linear.
   - No hidden layer within the network has fewer hidden neurons than either the input or output layer. This would rule out any problem with a single label per observation.

   Then, the results are derived within three even more restricted settings: (1) the network has exactly two layers (2) the network is learned upon a dataset consisting of a single example or (3) the data satisfies the RIP property. Some of the later theorems require even more assumptions such as $iid$ observations satisfying $\mathbb{E}_A \braket{A,M}^2 = ||M||^2_F$. I believe it is uncontroversial to say these results cannot be applied to any practical task. Therefore, I would challenge the authors to make the case as to how this work contributes itself or could have some downstream contribution to real-world problems/tasks. While this conference is a suitable venue for a theory-style paper such as this, I would argue that the choice of whether to accept such a paper should be heavily weighted by its potential practical impact (otherwise we can always invent interesting, but purely fictitious problems).

Minor issues/clarifications:
* While I realize that the experiment was straightforward, it is always nice to include the code used to generate the results. Could this be added to a camera-ready version?
* On L22+23 the paper states that "despite the overwhelming empirical evidence on ... the effectiveness of using sharpness regularization on improving generalization [15, 47, 51, 37], the connection between penalization of the sharpness of training loss and better generalization still remains unclear." Could the authors clarify how these are not the same thing? I suspect the point is clear to the authors but could just be worded more clearly.
* On L169: is this intended to refer to eqn (12)?
* Should the experiments in Fig 2 not also include a baseline of a model trained without label noise?



**Questions:**

* I found the terminology of "end-to-end parameters" (e.g. L52+53) to be slightly non-intuitive for its intended meaning. Possibly because in other parts of the machine learning literature this terminology refers to a different concept (models that are trained jointly in a non-modular way). Is this standard terminology in this literature? If not, it might be helpful to use an alternative or make this more clear.

**Limitations:**

Yes. I cannot foresee any potential negative societal impact of this work

---

> ### Author Rebuttal · Authors · 2023-08-09
>
> **"While this conference is a suitable venue for a theory-style paper such as this, I would argue that the choice of whether to accept such a paper should be heavily weighted by its potential practical impact (otherwise we can always invent interesting, but purely fictitious problems)"**
>
> **Reply**: Deep matrix factorization is a practically significant setting which was not invented in this work; it is established by previous Neurips papers, e.g., [Implicit regularization in matrix factorization, Gunasekar et al., NIPS 2017](https://proceedings.neurips.cc/paper_files/paper/2017/file/58191d2a914c6dae66371c9dcdc91b41-Paper.pdf), [Implicit regularization in deep matrix factorization, Arora et al., NIPS 2019](https://proceedings.neurips.cc/paper_files/paper/2020/file/f21e255f89e0f258accbe4e984eef486-Paper.pdf).
>
> **"The activations are linear."**
>
> **Reply**: As we pointed out, understanding the trace of Hessian regularizer for deep linear models is highly non-trivial as the regularizer is not computable in closed-form for depth beyond two. Therefore, similar to some of the previous impactful papers in the theory of deep learning, here we also focus on the first major difficult problem which is dealing with deep linear models.
>
> **"No hidden layer within the network has fewer hidden neurons than either the input or output layer. This would rule out any problem with a single label per observation."**
>
> **Reply**: This is not the case, for example if one chooses $A_i$ to be in the rank-one form $w^\top x_i$, for fixed weight vector $w$ and input vector $x_i$, then the single output prediction problem is recovered. The purpose of our work is to illustrate the surprising almost equivalence of the trace of hessian regularizer with the nuclear norm of the end-to-end matrix in deep matrix factorization, and the RIP condition is a natural and fairly general assumption on the data which allows the analysis to go through in a clean way without unnecessary lengthy derivations (indeed one can probably regenerate this result using some other concentration techniques but its beyond the purpose of this work.)
>
> ## Minor Issues and Questions:
> **It is always nice to include the code used to generate the results. Could this be added to a camera-ready version?**
>
>   **Reply**: Yes we will include the code in the camera-ready version.
>
> **On L22+23 the paper states that "despite the overwhelming empirical evidence on ... the effectiveness of using sharpness regularization on improving generalization [15, 47, 51, 37], the connection between penalization of the sharpness of training loss and better generalization still remains unclear." Could the authors clarify how these are not the same thing?**
>
>    **Reply**: Penalizing the sharpness of the training loss is an implicit bias mechanism, but it is not clear how this bias will impact the generalization behavior for general networks.
>
> **On L169: is this intended to refer to eqn (12)?**
>
>   **Reply**: Yes, but since it is a forward reference it might be clearer if we remove it, so we will rephrase this in the final version.
>
> **Should the experiments in Fig 2 not also include a baseline of a model trained without label noise?**
>
>   **Reply**: We will add the no label noise case to the final version, and also more settings. See the plots in the pdf included with this rebuttal.
>
> **I found the terminology of "end-to-end parameters" (e.g. L52+53) to be slightly non-intuitive for its intended meaning.**
>
>   **Reply**: We thank the reviewer for mentioning this delicate point, we will change this phrase to end-to-end matrix in the final version.

---

> > ### Comment · Reviewer_sgkx · 2023-08-11
> >
> > I thank the authors for their response.
> >
> > While I appreciate the references provided, I don't think this addresses the point I was making. Let me try to be more succinct.
> >
> > The results derived in this (and other similar) work(s) are under highly restrictive settings (linear data, linear activations, etc.) that don't represent the setting in which we are interested in better understanding the inductive biases of neural networks (e.g. typical architectures used in practice). Therefore, in works such as these, a vital consideration should be to what extent the results transfer over. Admittedly this might be something we can only investigate empirically but without doing so it is impossible for us to know if the intuitions one might obtain from theoretical work such as this should guide our intuitions in practical settings. Without doing so I would argue that this work is incomplete. I think this would be an obvious expectation on any empirical work in such a restrictive setting, therefore I think we should maintain the same standard for theoretical works.
> >
> > While I appreciate there is precedent for theoretically investigating restrictive settings, that does not in itself imply that we should continue deepening our research into these settings without consideration of the value of such results.

---

> > > ### Author Response · Authors · 2023-08-21
> > >
> > > Thank you for your response.
> > > Regarding the reviewer's response, we would like to briefly remind the following points:
> > >
> > > * Theoretical study is important in the long run.
> > > * Theoretical studies have to start with restricted cases even though there is a gap with practical world; many successful theoretical research started with linearized models; without understanding easier case, there is no chance to understand practical, complex cases.

---

### Author Rebuttal · Authors · 2023-08-09

We would like to thank the reviewers for their effort to provide helpful comments regarding our work. In the following, we have replied to their comments separately.

---

### Decision · Program_Chairs · 2023-09-21

**Decision:**

Accept (poster)

**Comment:**

Recent studies involving over-parameterized neural networks have illuminated the fact that the inherent stochastic nature of optimizers has an implicit regulatory impact on reducing the steepness of the loss function in particular as it relates to the trace of its Hessian matrix, particularly when considering solutions where the loss is reduced to zero. Additionally, there have been empirical observations indicating that more explicit variations of flatness regularization contribute to an enhancement in the generalization capabilities of these networks. This paper aims to study this typo of phenomenon for training of deep linear networks and shows under RIP minimizing the trace of the Hessian is approximately equivalent to minimizing the nuclear norm.

Most reviewers thought the paper was well written. They did raise concerns about the limitations to linear networks, the result not being directly about first order methods etc. One reviewer in particular expressed concerns about the practical relevance of these assumptions. Some of these concerns were alleviated in the discussion. My own reading of the paper is that is well written, it is a cute but not substantial or particularly hard result to prove from a theoretical standpoint. I also agree with the limitations raised by the reviewer. However, I still think the paper would be of interest as an initial study and merits publication in Neurips.